# Effect of exercise training on heart rate variability in type 2 diabetes mellitus patients: A systematic review and meta-analysis

Mathilde Picard[1], Igor Tauveron[2], Salwan Magdasy[2], Thomas Benichou[1], Reza Bagheri[3], Ukadike C. Ugbolue[4], Valentin Navel[5], Frédéric Dutheil[6]*

1 Endocrinology Diabetology and Metabolic Diseases, CHU Clermont–Ferrand, University Hospital of Clermont–Ferrand, Clermont-Ferrand, France, 2 Endocrinology Diabetology and Metabolic Diseases, Université Clermont Auvergne, GReD, CNRS, INSERM, CHU Clermont–Ferrand, University Hospital of Clermont–Ferrand, Clermont–Ferrand, France, 3 Exercise Physiology, University of Isfahan, Isfahan, Iran, 4 Health and Life Sciences, Institute for Clinical Exercise & Health Science, University of the West of Scotland, University of Strathclyde, Glasgow, Scotland, United Kingdom, 5 Translational Approach to Epithelial Injury and Repair, CHU Clermont-Ferrand, Université Clermont Auvergne, CNRS, INSERM, GReD, University Hospital of Clermont-Ferrand, Ophthalmology, Clermont-Ferrand, France, 6 Université Clermont Auvergne, CNRS, LaPSCo, Physiological and Psychosocial Stress, University Hospital of Clermont–Ferrand, CHU Clermont–Ferrand, Occupational and Environmental Medicine, WittyFit, Clermont–Ferrand, France

* frederic.dutheil@uca.fr

## Abstract

### Background

Cardiac autonomic neuropathy is a common complication of type 2 diabetes mellitus (T2DM), that can be measured through heart rate variability (HRV)–known to be decreased in T2DM. Physical exercise can improve HRV in healthy population, however results are under debate in T2DM. We conducted a systemic review and meta-analysis to assess the effects of physical exercise on HRV in T2DM patients.

### Method

PubMed, Cochrane, Embase, and ScienceDirect databases were searched for all studies reporting HRV parameters in T2DM patients before and after exercise training, until September 20th 2020, without limitation to specific years. We conducted random-effects meta-analysis stratified by type of exercise for each of the HRV parameters: RR–intervals (or Normal to Normal intervals–NN), standard deviation of RR intervals (SDNN), percentage of adjacent NN intervals varying by more than 50 milliseconds (pNN50), root mean square of successive RR-intervals differences (RMSSD), total power, Low Frequency (LF), High Frequency (HF) and LF/HF ratio. Sensitivity analyses were computed on studies with the highest quality.

### Results

We included 21 studies (9 were randomized) for a total of 523 T2DM patients: 472 had an exercise training and 151 were controls (no exercise). Intervention was endurance (14 studies), resistance (2 studies), endurance combined with resistance (4 studies), and high intensity interval training (HIIT) (4 studies). After exercise training, all HRV parameters improved

**Data Availability Statement:** All relevant data are included within this article and its Supporting information.

**Funding:** The authors received no specific funding for this work.

**Competing interests:** The authors have declared that no competing interests exist.

i.e. an increase in SDNN (effect size = 0.59, 95%CI 0.26 to 0.93), RMSSD (0.62, 0.28 to 0.95), pNN50 (0.62, 0.23 to 1.00), HF (0.58, -0.16 to 0.99), and a decrease in LF (-0.37, -0.69 to -0.05) and LF/HF (-0.52, -0.79 to -0.24). There were no changes in controls. Stratification by type of exercise showed an improvement in most HRV parameters (SDNN, RMSSD, pNN50, LF, HF, LF/HF) after endurance training, whereas mostly LF/HF was improved after both resistance training and HIIT. Supervised training improved most HRV parameters. Duration and frequency of training did not influence the benefits on HRV.

## Conclusion

Exercise training improved HRV parameters in T2DM patients which may reflect an improvement in the activity of the autonomic nervous system. The level of proof is the highest for endurance training. Supervised training seemed beneficial.

## 1. Introduction

Type 2 diabetes mellitus (T2DM), a multifactorial metabolic disorder, has become a global epidemic with a worldwide increasing prevalence [1]. Although there are more than 400 million people with T2DM, by 2045 the prevalence is projected t to increase by 51% [1, 2]. Among the many complications of T2DM, cardiac autonomic neuropathy (CAN) is one of the most serious, being strongly associated with the risk of mortality [3]. Its development is associated with the lesion of the autonomic nervous system and may be accompanied by coronary vessel ischemia, arrhythmias, "silent" myocardial infarction, severe orthostatic hypotension, and sudden death syndrome [4].

Interestingly, CAN can be measured through heart rate variability (HRV), that is strongly decreased in T2DM [5, 6]. Despite the gold standard to assess CAN using cardiovascular reflex tests [7], one of the most convenient and reliable assessments is through HRV. HRV can be measured easily using a portable device, non–intrusively and pain–free [8]. The autonomous system degeneration may occur quite early in the course of diabetes and the HRV analysis could be used for detecting subclinical CAN even before demonstration of clinical sign and symptoms. The HRV analysis can provide detailed information about the cardiac regulatory system and it has been demonstrated that T2DM patients exhibit a strong decrease in HRV [5, 6]. HRV is basically the variation between two consecutive heartbeats (RR-intervals) [9]. HRV can be analyzed through various parameters, classically classified as time and frequency domains. Time domains are calculation from RR-intervals (time between two heartbeats), and frequency domains are a more complex power spectral analysis of the HRV. Both domains comprise several parameters that provide information on the activity of the autonomic nervous system, such as sympathetic or parasympathetic activity [10].

Intensified multifactorial intervention in patients with T2DM reduced the risk of CAN progression by 68% [11–13]. Lifestyle modifications with increased physical activity and structured exercises can lead to improvements in HRV, independently of weight change in persons at high risk for diabetes, and in patients with T2DM [11, 14]. Exercise training is a cornerstone of lifestyle intervention [15–17], leading to improved HRV in healthy population [18], but it remains unclear to what extent physical exercise can improve HRV in T2DM. In T2DM, different modalities of exercise have been tested such as endurance [19–21], resistance [22, 23], or high intensity interval training (HIIT) [24, 25]. HIIT provides greater benefits to functional capacity compared to endurance training [26]. Resistance training likewise endurance

training, improves metabolic features, insulin sensitivity and reduces abdominal fat [27, 28]. However, benefits on HRV depending on exercise modality remain unclear. Moreover, other characteristics of training may influence the results. For example, supervised exercises have been proven to be more effective than non-supervised exercises, based on several outcomes [29–31]. Similarly, duration and frequency of training, are strongly linked with putative benefits, but evidence is scarce on HRV in T2DM. Characteristics of patients can also influence benefits of exercise on HRV [32, 33]. Lastly, the relationships between changes in HRV and clinical or biological parameters has also been poorly studied [34–36].

Therefore, we aimed to conduct a systematic review and metanalysis 1) on the impact of exercise on HRV in patients with T2DM, 2) depending on modalities of exercise such as the type of exercise, its supervision or not, or duration and frequency of sessions, 3) and depending on characteristics of patients.

## 2. Methods

### 2.1 Ethics statement

Ethics approval and consent to participate were not applicable for a systematic review and meta-analysis. We did not include personal data or patients in this systematic review and meta-analysis. All authors have agreed to publish the results of this work.

### 2.2 Literature search

We reviewed all studies reporting the effect of exercise training on HRV in T2DM patients. Animal studies were excluded. The PubMed, Cochrane Library, Science Direct and Embase databases were searched until September 20th 2020, with the following keywords: diabetes AND (exercise OR physical) AND ("heart rate variability" OR HRV). The search was not limited to specific years and no language restrictions were applied. To be included, studies needed to describe our primary outcome variables i.e. HRV data before and after exercise training in T2DM patients, with or without a control group (no physical activity intervention). We excluded studies that assessed the effects of other intervention (such as dietary or psychological intervention) in combination with exercise training. Conferences, congresses or seminars, were excluded. In addition, reference lists from all publications meeting the inclusion criteria were manually searched to identify any further studies not found through the electronic search. Ancestry searches were also completed on previous reviews to locate other potentially eligible primary studies. Two authors (Mathilde Picard, Dutheil Frédéric) conducted the literature searches, reviewed the abstracts, and based on the selection criteria, decided the suitability of the articles for inclusion, and extracted the data. When necessary, disagreements were solved with a third author (Valentin Navel). We followed the guidelines outlined by PRISMA [37] (S1 Checklist).

### 2.3 Data extraction

The primary outcome analysed was HRV parameters. Time-domain parameters were RR–intervals (or Normal to Normal intervals–NN), standard deviation of RR intervals (SDNN), percentage of adjacent NN intervals varying by more than 50 milliseconds (pNN50), and root mean square of successive RR-intervals differences (RMSSD). Frequency-domain parameters were total power (TP), low frequency (LF), high frequency (HF) and LF/HF ratio. The RMSSD and pNN50 are associated with HF power and hence parasympathetic activity, whereas SDNN is correlated with LF power. Although LF power is an index of both sympathetic and parasympathetic activity, LF power is commonly considered as a measure of sympathetic modulations,

**Table 1. Descriptive characteristics of HRV parameters.**

| HRV parameters | | | |
|---|---|---|---|
| Acronym | Full name | Unit | Interpretation |
| **Time-domain** | | | |
| RR | RR–intervals (or Normal to Normal intervals–NN) i.e. beat-by-beat variations of heart rate | ms | Overall autonomic activity |
| SDNN | Standard deviation of RR intervals | ms | Correlated with LF power |
| RMSSD | Root mean square of successive RR-intervals differences | ms | Associated with HF power and hence parasympathetic activity |
| pNN50 | Percentage of adjacent NN intervals varying by more than 50 milliseconds | % | Associated with HF power and hence parasympathetic activity |
| **Frequency-domain** | | | |
| TP | Total power i.e. power of all spectral bands | $ms^2$ | Overall autonomic activity |
| LF | Power of the high-frequency band (0.15–0.4 Hz) | $ms^2$ (absolute power) or nu (relative power in normalized unit) | Index of both sympathetic and parasympathetic activity, with a predominance of sympathetic |
| HF | Power of the high-frequency band (0.15–0.4 Hz) | | Represents the most efferent vagal (parasympathetic) activity to the sinus node |
| LF/HF | LF/HF ratio | - | Sympathovagal balance |

particularly when expressed in normalised units. In practical terms, an increase of the LF component is generally considered to be a consequence of an increased sympathetic activity [38]. HF power represents the most efferent vagal (parasympathetic) activity to the sinus node [8, 39–42]. Therefore, an increase of the HF component reflects an increased parasympathetic activity. The LF/HF ratio represents the sympathovagal balance (Table 1). Secondary outcomes included HRV parameters before exercise training, characteristics of training (modalities of exercise such as endurance, resistance, or high intensity interval training; supervised or not; duration and frequency of sessions; duration of training; intensity), characteristics of T2DM (duration of T2DM, HbA1c, treatments), clinical (body mass index, blood pressure, $VO_2$max or $VO_2$peak, treatments) and biological (total cholesterol, triglycerides, LDL-cholesterol, HDL-cholesterol) parameters, and sociodemographic (age, sex, smoking).

## 2.4 Quality of assessment

We used the Scottish Intercollegiate Guidelines Network (SIGN) criteria to check the quality of included articles, both for randomized and non-randomized clinical trials, with the dedicated evaluation grids. Checklists consisted of 10 and 7 items, respectively. We gave a general quality score for each included study based on the main causes of bias evaluated in section 1 of the checklist through 4 possibilities of answers (yes, no, can't say or not applicable) [43] (S1 Appendix). In addition, we also used the 0–10 Physiotherapy Evidence Database (PEDro) scale for a complementary overview of the quality of the studies (S2 Appendix).

## 2.5 Statistical considerations

We conducted meta–analyses on the effect of exercise on HRV parameters in T2DM. P values less than 0.05 were considered statistically significant. For the statistical analysis, we used Stata software (version 16, StataCorp, College Station, US) [44–48]. Main characteristics were synthetized for each study population and reported as mean ± standard-deviation (SD) for continuous variables and number (%) for categorical variables. We conducted random effects meta–analyses (DerSimonian and Laird approach) when data could be pooled (more than five data for the same outcome) [49]. Particular attention was paid towards short recordings (1 minute) of HRV parameters [50]. First, we calculated the effect size (ES, standardised mean

differences–SMD) [51] of each HRV parameter after exercise compared to baseline (before exercise) in T2DM. ES is a unitless measure centered at zero if HRV does not differ between measures before and after exercise. A positive ES denotes higher levels of the tested HRV parameter in T2DM patients after exercise. An ES of 0.8 reflects a large effect, 0.5 a moderate effect, and 0.2 a small effect. As ES is a unitless measure and as we compared data after and before exercise, frequency-domain HRV parameters measured in $ms^2$ or in normalized unit (nu) were combined. In addition, we conducted meta-analyses stratified on type of exercise (endurance, resistance, mixed, HIIT), supervision of exercise or not. We also conducted a meta-analysis on controls to verify the absence of changes within each HRV parameter. We searched for potential publication bias using funnel plots of all aforementioned meta–analyses and we evaluated heterogeneity by examining forest plots, confidence intervals (CI) and I-squared ($I^2$). A low heterogeneity is reflected by $I^2$ values <25%, modest for 25–50%, and high for >50%. We verified the strength of our results by conducting further meta–analyses (sensitivity analyses) after exclusion of studies that were not evenly distributed around the base of the funnel. Lastly, we reperformed the aforementioned meta-analyses only using the studies with the highest level of proof, i.e. only on randomised studies and only on randomized controlled studies. When possible (sufficient sample size), meta–regressions were proposed to study the relationship between changes in HRV parameter (RR intervals, RMSSD, pNN50, SDNN, total power, LF, HF, LF/HF) and clinically relevant parameters such as characteristics of intervention (type of exercise, supervised or not, duration and number of sessions, frequency, intensity), clinical parameters (time from T2DM diagnosis, HbA1c, etc.), sociodemographic (age, sex, etc.), or methods of measurement of HRV, and their changes when pertinent. Results were expressed as regression coefficients and 95% CI.

## 3. Results

An initial search produced 6641 possible articles. After removal of duplicates using Zotero® software, all possible articles were manually checked by two authors. The use of the selection criteria reduced the number of articles reporting the effect of exercise on HRV in T2DM patients to 21 articles in the systematic review, among which 18 articles were included in the meta-analysis (inter-reader agreement κ = 0.89) (Fig 1). Three articles were not included in the meta-analysis because they only reported ratio between longest and shortest RR-intervals; they were also distinguishable because they measured HRV manually on electrocardiogram recording of 1 minute [19, 52, 53]. All included articles were written in English. The main characteristics of the studies are reported in Table 2.

### 3.1 Quality of articles

The assessment of the quality of these 21 studies was performed using the score SIGN. Results varying from 20% [24] to 80 [23] for Yes responses, with a mean score of 45.7 ± 14.7. Few studies showed a high level of proof mainly due to the lack of a control group or a poor method of randomisation (S2 Appendix and S1 Fig). Using the PEDro scale, scores ranged from 2 [19, 20, 24] to 6 [23, 54–56] out of 10 (S2 Appendix and S2 Fig).

### 3.2 Study designs and objectives

Included studies were published from 2003 to 2019 and conducted in various geographic locations. All the 21 included studies measured HRV parameters in T2DM patients before and after physical exercise program. Seven studies had a control group of T2DM patients without exercise: five were RCT [25, 52, 54, 56, 57] and two were non-RCT [19, 58]. Five studies compared different T2DM groups of exercise (and had no group without exercise): four were

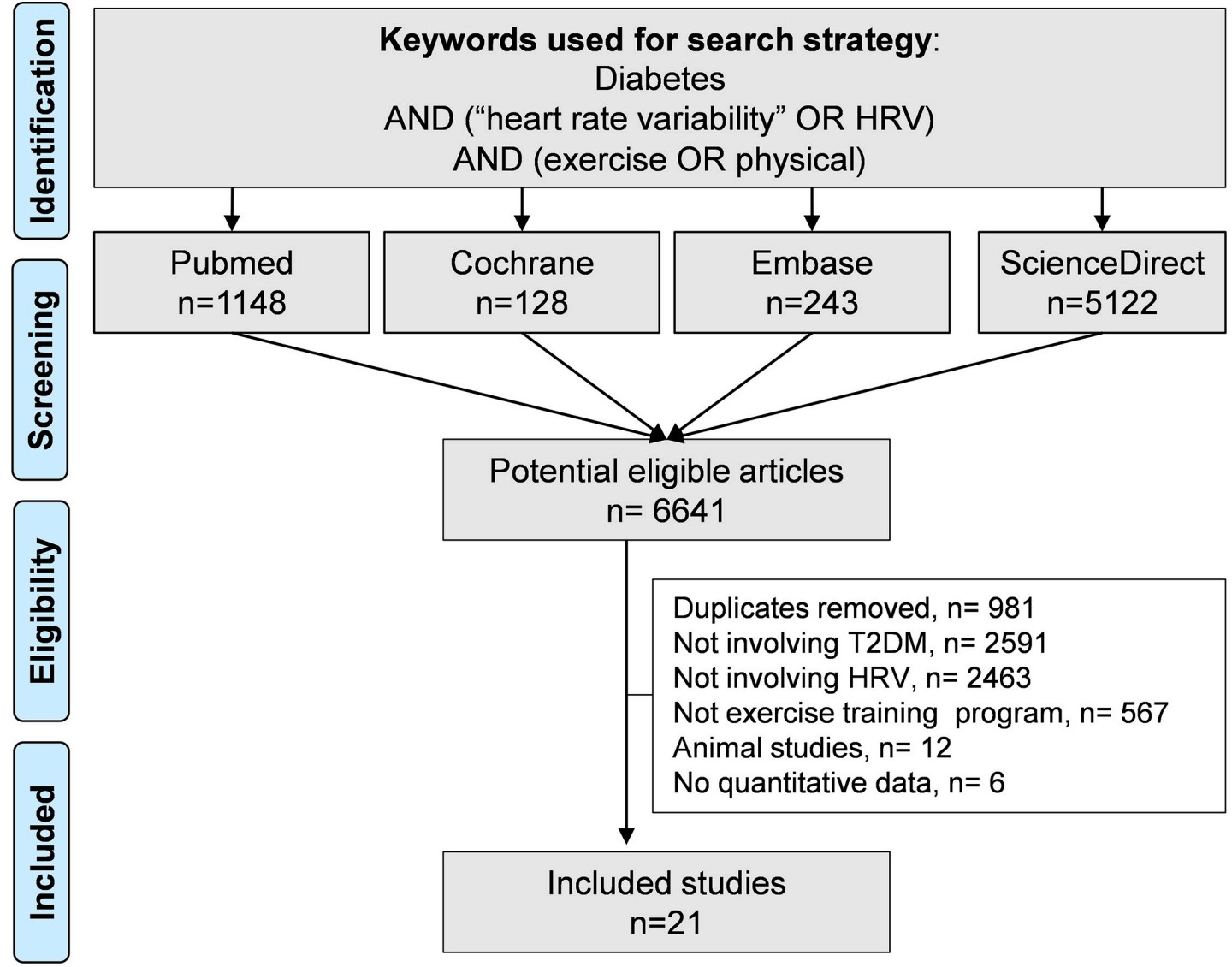

**Fig 1. Flow diagram in accordance with the PRISMA guidelines.**

randomized [23, 53, 55, 59], and one not [22]. They assessed the influence of the type [23, 53, 55] or frequency/duration [22, 59] of exercise. Four studies had a control group of non-T2DM patients undergoing the same exercise training than the T2DM [20, 21, 60, 61]. One study compared T2DM patients with or without cardiac autonomic neuropathy [62]. Four were pre-post single group studies [24, 63–65].

### 3.3 Study characteristics: Inclusion and exclusion criteria

Included patients had to have T2DM, without further details for most studies. Inclusion criteria for TD2M patients were biological in five studies (fasting glucose > 126 mg/dl [19, 22, 52, 60, 61] or glucose levels > 200 mg/dl after an oral glucose tolerance test [60, 61]). Most studies included patients with sedentary behavior or low level of physical activity [21–23, 56, 60–62, 65, 66], i.e. less than 3 hours of physical activity per week [65] or less than 60 min moderate

**Table 2. Characteristics of included studies.**

| Study | Country | Design | Patients | | | Type / group | Duration (months) | Exercise | | | Supervision | HRV measures | | | Other outcomes* |
|---|---|---|---|---|---|---|---|---|---|---|---|---|---|---|---|
| | | | n analyzed | Age, mean (years ±SD) | Men (%) | | | Intensity | Session | /week | | HRV parameters* | Deep breathing | ECG (min) | |
| Abdelbasset 2019 | Egypt | RCT | 20 | 52.4 ±4.6 | 100% | HIIT | 4 | 4 intervals of 4 min at 80–90% of MHR with 2 min at 50–60% of MHR between each | 30 min | 3 | Yes | HRV lying (RR↘, SDNN↗, RMSSD↗, LF, HF↗, LF/HF↘) | No | 5 | HbA1c, BMI, total cholesterol, HDL, LDL, triglycerides, VO2 |
| | | | 20 | 51.5 ±5.1 | | Control | | | | | - | | | | |
| Bellavere 2018 | Italy | R Comp. Type of exercise | 19 | 57.1 ±1.6 | 68% | End. | 4 | Up to 60–65% of HRR | 60 min | 3 | Yes | HRV lying and standing (TP, LF↘, HF↗, LF/HF↘) | No | 10 | HbA1c, BMI, total cholesterol, HDL, LDL, triglycerides, VO2 |
| | | | 11 | 53.4 ±2.0 | 82% | Resist. | | Up to 3 sets of 10 repetitions at 70–80% 1-RM | | | Yes | | | | |
| Bhagyalakshmi 2007† | India | Non-RCT | 28 | 61.8 ±3.1 | 79% | End. | 9 | Unclear | 45 min | 7 | Yes | HRV lying (↗ difference shortest and longest RR) | Yes | 1 | HbA1c |
| | | | 20 | 59.5 ±2.8 | 50% | Control | | | | | - | | | | |
| Cassidy 2019 | UK | RCT | 11 | 60.0 ±3.0 | 82% | HIIT | 3 | 5 intervals of 2 min (up to 3 min 50 s) at > 80 RPM and RPE 16–17 with 3 min recovery between each | 30–40 min | 3 | No | HRV lying (RR, SDNN, LF, HF, LF/HF) | No | 20 | HbA1c, BMI, blood pressure, VO2 |
| | | | 11 | 59.0 ±3.0 | 73% | Control | | | | | - | | | | |
| Duennwald 2014 | Austria | R Comp. Type of exercise | 8 | 59.6 ±5.7 | 75% | HIIT | 1 | 5 intervals of 4 min at 90–95% of MHR with 3 min at 70% of MHR between each 70% of MHR | 42 min | 3 | Yes | HRV lying (RR, SDNN) | No | 4 | HbA1c, BMI, blood pressure, total cholesterol, HDL, VO2 |
| | | | 7 | 59.6 ±6.1 | 71% | End. | | | 50.3 min | | Yes | | | | |
| Faulkner 2014 | USA | Non-R T2DM vs no T2DM | 9 | 14.7 ±1.8 | 11% | End. (T2DM) | 4 | 60–75% of MHR | 60 min | 5 | No | HRV 24h (RMSSD, pNN50, SDNN, TP, LF, HF) | No | 1440 | HbA1c, BMI, total cholesterol, HDL, LDL, triglycerides, VO2 |
| | | | 10 | 14.6 ±1.6 | 40% | End. (No T2DM) | | | | | No | | | | |
| Figueroa 2007 | USA | Non-R T2DM vs no T2DM | 8 | 50.0 ±2.8 | 0% | End. (T2DM) | 4 | 65% of VO2peak | 30–45 min | 4 | +/- | HRV lying (LF, HF, LF/HF) | Yes | 5 | HbA1c, BMI, blood pressure, VO2 |
| | | | 12 | 48.0 ±6.9 | 0% | End. (No T2DM) | | | | | +/- | | | | |

(Continued)

Table 2. (Continued)

| Study | Country | Design | Patients | | | Exercise | | | | | | HRV measures | | | Other outcomes* |
|---|---|---|---|---|---|---|---|---|---|---|---|---|---|---|---|
| | | | n analyzed | Age, mean (years ±SD) | Men (%) | Type / group | Duration (months) | Intensity | Session | /week | Super-vision | HRV parameters* | Deep breathing | ECG (min) | |
| Goit 2014 | Nepal | Pre-post | 20 | 42.2 ±6.4 | 100% | End. | 6 | 60–85% of HRR | 50 min | 3 | Yes | HRV lying (SDNN, RMSSD↗, pNN50↗, LF↘, HF↗, LF/HF↘) | No | 5 | HbA1c, BMI, blood pressure, HDL, LDL, triglycerides |
| Goit 2017 | Nepal | Pre-post | 41 | 44.2 ±4.5 | 100% | End. | 6 | 50–70% of HRR | 50 min | 3 | Yes | HRV lying (SDNN↗, RMSSD↗, pNN50↗, LF↘, HF↗, LF/HF↗) | No | 5 | HbA1c, BMI, blood pressure, total cholesterol, HDL, LDL, triglycerides |
| Gouloupoulou 2010 | USA | Non-R. T2DM vs no T2DM | 26 | 50.0 ±5.1 | 50% | End. (T2DM) | 4 | 65% of VO₂peak | 30–45 min | 4 | +/- | HRV lying (TP) | Yes | 5 | HbA1c, BMI, blood pressure, total cholesterol, HDL, LDL, triglycerides, VO₂ |
| | | | 36 | 49.0 ±6.0 | 39% | End. (No T2DM) | | | | | +/- | | | | |
| Kanaley 2009 | USA | Non-R. T2DM vs no T2DM | 22 | 50.0 ±1.6 | 45% | End. (T2DM) | 4 | 65% of VO₂peak | 30–45 min | 4 | +/- | HRV lying (TP) | Yes | 5 | HbA1c, BMI, VO₂ |
| | | | 34 | 49.0 ±0.9 | 38% | End. (No T2DM) | | | | | | | | | |
| Kang 2016 | Korea | RCT | 8 | 56.0 ±7.4 | 0% | End. & Resist. | 3 | 60% of HRR + 2 sets 9 exercises 8–12 repetitions at 1-RM of 60–80% | 30 + 30 min | 3 | Yes | HRV lying (RMSSD, SDNN, LF, HF, LF/HF) | No | 5 | HbA1c, BMI, blood pressure, total cholesterol, HDL, LDL, triglycerides, VO₂ |
| | | | 8 | 57.5 ±4.6 | 0% | Control | | | | | - | | | | |
| Loimaala 2003 | Finland | RCT | 24 | 53.6 ±6.2 | 100% | End. & Resist. | 12 | 65–75% of VO₂peak 3 sets of 10–12 repetitions at 70–80% max. voluntary contraction | 30 min | 4 | +/- | HRV lying (SDNN, pNN50, LF, HF, LF/HF) | No | 1440 | HbA1c, BMI, blood pressure, VO₂ |
| | | | 25 | 54.0 ±5.0 | | Control | | | | | - | | | | |
| Moawd 2015 | Egypt | Non-R Comp. Type of exercise | 20 | 50.5 ±8.6 | 100% | End. | 3 | 60–85% of HRR RPE "very to fairly light" up to "somewhat hard" | 50 min | 3 | Yes | HRV lying (RMSSD↗, SDNN↗, pNN50↗) | No | 10 | BMI, blood pressure |
| | | | 18 | 51.3 ±6.1 | 100% | Resist. | | | 5–20 min | | - | | | | |

(Continued)

**Table 2.** (Continued)

| Study | Country | Design | Patients | | | Exercise | | | | | | HRV measures | | | Other outcomes* |
|---|---|---|---|---|---|---|---|---|---|---|---|---|---|---|---|
| | | | n analyzed | Age, mean (years ±SD) | Men (%) | Type/group | Duration (months) | Intensity | Session | /week | Super-vision | HRV parameters* | Deep breathing | ECG (min) | |
| **Pagkalos 2008** | Greece | Non-R Comp. CAN status | 15 | 55.8 ±5.6 | 27% | End. (CAN -) | 6 | 70–85% of HRR | 45–75 min | 3 | Yes | HRV lying (RMSSD↘, SDNN↗, pNN50↗, LF↘, HF↗, LF/HF↘) | No | 1440 | HbA1c, BMI, blood pressure, total cholesterol, HDL, LDL, triglycerides, VO₂ |
| | | | 17 | 56.2 ±5.8 | 35% | End. (CAN +) | | | | | Yes | | | | |
| **Parpa 2009** | USA | Pre-post | 14 | 57.0 ±6.7 | 36% | HIIT | 3 | 6 intervals of 2 min at 80–90% of MHR with 2 min at 50–60% MHR between each | 30 min | 4 | Yes | HRV lying (SDNN↗) | No | 5 | Blood pressure |
| **Sacre 2014** | Australia | Non-RCT | 22 | 59.0 ±10.0 | 59% | End. & Resist. | 6 | RPE "moderate-vigorous" | 75 min | 2 | +/- | HRV lying (RR↘, SDNN↗, RMSSD, TP↗, LF, HF, LF/HF) | No | 5 | HbA1c, BMI, blood pressure, total cholesterol, HDL, LDL, triglycerides |
| | | | 25 | 60.0 ±9.0 | 40% | Control | | | | | | | | | |
| **Simmonds 2012** | Australia | R Comp. frequency & duration of sessions | 8 | 68.6 ±2.8 | 0% | End. | 3 | At 100% of Tge | 60 min | 2 | unclear | HRV lying (RR, SDNN↘, RMSSD↗, LF, HF↗, LF/HF) | No | 10 | HbA1c, total cholesterol, HDL, LDL, triglycerides, VO₂ |
| | | | 7 | 69.3 ±2.5 | 0% | End. | | | 30 min | 4 | unclear | | | | |
| **Sridhar 2010[†]** | India | RCT | 55 | 61.8 ±3.1 | 56% | End. | 12 | Unclear | 45 min | 5 | Yes | HRV lying: ratio between longest RR and shortest | Yes | 1 | HbA1c, BMI, blood pressure |
| | | | 50 | 59.5 ±2.8 | 55% | Control | | | | | - | | | | |
| **Wormgoor 2018[†]** | New Zealand | R Comp. Type of exercise | 11 | 52.5 ±7.0 | 100% | End. & Resist. | 3 and 6 | Up to 26 min 55% eWLmax & 2 sets of 12 repetitions at 75% of 1-RM | 60 min | 3 | +/- | HRV lying: ratio between longest RR and shortest | Yes | 1 | HbA1c, BMI, HDL, triglycerides, VO₂ |
| | | | 11 | 52.2 ±7.1 | 100% | HIIT & Resist | | Up to 12 (or 8) intervals of 1 min at 95% (or 120) eWLmax with 1 (or 2.25) min at 40 (or 30)% eWLmax between each & 2 sets of 12 repetitions at 75% of 1-RM | | | +/- | | | | |

(Continued)

**Table 2.** (Continued)

| Study | Country | Design | Patients | | | Exercise | | | | | | HRV measures | | | Other outcomes* |
| --- | --- | --- | --- | --- | --- | --- | --- | --- | --- | --- | --- | --- | --- | --- | --- |
| | | | n analyzed | Age, mean (years ±SD) | Men (%) | Type / group | Duration (months) | Intensity | Session | /week | Super-vision | HRV parameters* | Deep breathing | ECG (min) | |
| **Zoppini 2007** | Italy | Pre-post | 12 | 65.7 ±5.6 | 42% | End. | 6 | 50–70% of HRR | 60 min | 2 | Yes | HRV lying (RR, TP, LF, HF, LF/HF) | No | 10 | HbA1c, BMI, blood pressure, HDL, LDL, triglycerides |

1-RM: one repetition maximum, DB: Deep Breathing, End.: endurance training, eWLmax: estimate maximal workload, HBP: high-blood pressure; HIIT: high intensity interval training, HRR: heart rate reserve, MHR: maximum heart rate, N: sample size, Non-RCT: non-randomized controlled trial, R Comp. Type of exercise: randomized comparative on type of exercise, RCT: randomized controlled trial, Resist.: resistance training, RPE: rating of perceived exertion, RPM: revolutions per minute, Tge: gas-exchange threshold, +/- means partially supervised, * supervised during 12 weeks, not supervised after, data at 6 months considered as "partially supervised".

*: only significant increase (↗) or decrease (↘) are presented (differences between groups or after vs before exercise training).

†: not included in the meta-analyses.

vigorous activity per week [25], or not being involved in regular physical activity [20–23, 60–62]. Some studies required patients to be aged over 35 [53], 40 [23, 58, 60, 61] or 65 [59] years old, or under 60 [53, 60, 61], 65 [55], 70 [23], 74 [59] years old, or between 12 and 19 years old [20]. Nine studies included sex-specific populations: men [22, 53, 56, 57, 63, 64] or women [21, 54, 59]. Six studies included patients according to their body mass index (BMI): >30 [21, 56, 60, 61, 64] or between 24 and 36 kg.m$^{-2}$ [23]. The main exclusion criteria were: smoking [21, 23, 24, 55, 56, 59–61, 63–65], exogenous insulin [21, 22, 25, 55, 56, 59, 63], beta blockers or arrythmia [19, 21, 23, 25, 52, 55, 56, 59–61, 63, 64], and cardiovascular disease [21, 23–25, 52, 53, 55, 56, 58, 60, 61, 63–66].

## 3.4 Characteristics of population

**3.4.1 Sample size.** Ranged from 11 [65] to 105 [52]. We included a total of 623 T2DM patients: 472 underwent an exercise training, and 151 were controls (no exercise training).

**3.4.2 Age.** The mean age of T2DM patients following exercise training was 54.5 years (95% CI 48.6 to 60.4), ranging from 69.3 ±2.5 [59] to 14.7 ±1.8 [20], and 58.4 years (95% CI 56.0 to 60.9) in the T2D controls, ranging from 60 ±9 [58] to 51.8 ±5.1 [56].

**3.4.3 Gender.** The proportion of men varied from 0% [21, 54, 59] to 100% [22, 53, 56, 57, 63, 64] in T2DM patients following exercise training, and also from 0% [54] to 100% [56, 57] in T2DM not following any exercise training with a mean of 60% (95% CI 50 to 70) and 60% (95% CI 36 to 83) respectively.

**3.4.4 T2DM duration.** The mean time from T2DM diagnosis was 9.1 years (95% CI 7.0 to 11.1) ranging from 18.6 ±4.6 [63] to 1.6 ±1.4 [20] years for T2D patients following exercise training and 6.2 years (95% CI 5.0 to 7.4) ranging from 8.3 ±4.2 [56] to 5 ±1 [25] for T2D controls. T2DM duration was not reported in 7 studies [21, 24, 54, 57, 59–61].

**3.4.5 Metabolic control.** (HbA1c) was reported in all studies except two [22, 24]. The mean HbA1c patients following exercise training was 7.5% (95% CI 7.2 to 7.7) in T2DM patients, ranging from 10.4 ±2.2 [64] to 6.4 ±0.6 [54], and 7.7% (95% CI 7.1 to 8.4) in controls, ranging from 8.7 ±0.32 [52] to 6.4 ±0.5 [54].

**3.4.6 BMI.** Was reported in all studies except four [19, 24, 59]. The mean BMI patients following exercise training was 29.5 kg/m$^2$ (95% CI 28.3 to 30.7) in T2DM patients, ranging from 39.2 ±9.4 [53] to 23.9 ±2.9 [54], and 28.3 kg/m$^2$ (95% CI 27.1 to 29.6) in controls, ranging from 34.6 ±1.8 [56] to 25.5 ±3.1 [54].

**3.4.7 Blood pressure.** was reported in all studies except seven [19, 20, 23, 53, 56, 59, 61]. Mean blood pressure (systolic/diastolic) following exercise training was 129.2/81.2 mmHg (95% CI 123.0/78.7 to 135.3/83.7) in T2DM patients, and ranged from 144.2/88.6 [52] to 117.3 [62] / 61.5 [60]. The Mean blood pressure was 133.2/79.6 mmHg (95% CI 119.8/73.0 to 146.7/86.1) in the control group without exercise, and ranged from 145.2/87.0 [52] to 119.0 [25] / 70.0 [58].

**3.4.8 Blood lipid levels.** Total cholesterol was reported in 11 studies [20, 23, 54–56, 58–60, 62–64] HDL cholesterol in 13 studies [20, 23, 53–56, 58–60, 62, 63, 65, 67], LDL in 11 [20, 23, 54, 56, 58–60, 62–65] and triglycerides in 12 studies [20, 23, 53, 54, 56, 59, 60, 62–65, 68].

**3.4.9 Aerobic capacity.** VO$_2$max and VO$_2$peak were reported in 3 [53, 54, 57] and 10 studies [20, 21, 23, 25, 55, 56, 59–62] respectively. Most studies measured VO$_2$ with gas exchange analysis during incremental exercise tests using cycle ergometer [20, 23, 25, 53, 55] or treadmill [21, 56, 57, 59–62].VO$_2$max was extrapolated from sub-maximal measures in one study [54]. We chose to use VO$_2$peak as a generalization in this article. Mean VO$_2$peak was 24.3 mL.min$^{-1}$.kg$^{-1}$ (95% CI 22.3 to 26.2) for exercise groups; before commencing the training program, VO$_2$peak ranged from 18 ±2.8 mL.min$^{-1}$.kg$^{-1}$ [59] to 31.9 ±5.1 [57]. Mean VO$_2$peak

was 25.2 mL.min$^{-1}$.kg$^{-1}$ (95% CI 20.5 to 29.9) for controls. and ranged from 20.3 ±1.8 [25] to 32.2 ±6.4 [57].

## 3.5 Intervention

**3.5.1 Type of exercise.** Most studies (18/21) explored the effects of endurance training. Among those, 4 studies had endurance combined with resistance training [53, 54, 57, 58]. Two studies had only a group of resistance training [22, 23]. Five studies had an HIIT intervention [24, 25, 53, 55, 56], in combined form with resistance training in one study [53]. For endurance training and HIIT, treadmill or cycle ergometer were used in most of the studies, and outdoors walking in others, without further details in two study [57, 58]; stepper seat [22], isotonic machine [53] or weigh machine [23, 54] were used for resistance exercises.

**3.5.2 Duration of exercise session.** The duration of each exercise session ranged from 30 [21, 24, 25, 56, 57, 59–61] to 75 min [58, 62]. The duration gradually increased during the training program for 7 studies [21, 22, 25, 60–62, 68].

**3.5.3 Frequency.** The frequency of exercise session ranged from 2 [57, 58, 65] to 7 [19] times per weeks; 3 times per week in ten studies [22, 23, 25, 53–56, 62–64], 4 times in four studies [21, 24, 60, 61], 5 times in two studies [20, 52], and one study compared the effect of exercise twice and 4 times a week [59].

**3.5.4 Duration of intervention.** Varied from 1 [55] to 12 [52, 57] months; 3 months in five studies [22, 24, 25, 54, 59], 4 months in 6 studies [20, 21, 23, 56, 60, 61], 6 months in 6 studies [53, 62–65, 68] and 9 months in one study [19].

**3.5.5 Intensity.** For the endurance training program, the targeted intensity was based on maximal heart rate for 2 studies [20, 55] (varying from 60% [20] to 70% [55]), on heart rate reserve for 7 studies [22, 23, 54, 62–65] (varying from 50–70% [64, 65] to 70–85% [62]), and for percentage of VO$_2$peak in 4 studies [21, 57, 60, 61] (varying from 65% studies [21, 60, 61] to 65–75% [57]). Two studies did not provide precise details regarding the intensity of exercises [19, 52] and one used "moderate vigorous" exercises without further details [58]. The control of the intensity during exercise was achieved by using a heart rate monitor in 13 studies [20, 21, 23, 24, 53–55, 57–62], the Borg scale in one study [25], and was not reported in 7 studies [19, 22, 52, 56, 63–65]. Heterogeneity of intensity measurements precluded further analyses.

**3.5.6 Supervision.** The exercises were supervised in 12 studies [19, 22–24, 52, 54–56, 62–65], in one study the exercises were supervised for 3 months and non-supervised for 3 additional months [53], in 5 studies exercises were partially supervised [21, 57, 58, 60, 61], in 2 studies exercise were not supervised [20, 25], and the supervision was not mentioned in one study [59].

## 3.6 HRV measures

**3.6.1 Measures condition.** Most studies used ECG, achieved in a resting supine position, to determine HRV [19, 21, 23, 25, 53, 55, 56, 58, 60, 61, 63, 65, 67] up to 20 minutes [25]. Two studies used a 24-hour holter-ECG [20, 62] and two other studies used a chest strap coupled to a wristwatch receiver (Polar Electro Oy) [57, 59]. Most studies measured HRV at spontaneous breathing, while in deep breathing in six studies [19, 21, 52, 53, 60, 61]. After recording, most studies used a data acquisition system, except three that measured R-R intervals manually [19, 52, 63].

**3.6.2 Duration of measures.** Recordings lasted between one [19, 52, 53] and 20 min [25], 5 min in the majority of studies [21, 24, 54, 56, 60, 61, 63, 64, 68]; and 24 h in 3 studies [20, 57, 62].

**3.6.3 Parameters reported.** Most studies reported time and frequency domains, except three reporting only the ratio or the difference between the longest and the shortest RR interval [19, 52, 53]. For time domain parameters, RR intervals (RRI) was reported in 5 studies [25, 55, 59, 65, 68], SDNN in 13 studies [22, 24, 25, 54–57, 59, 62–64, 66, 68], RMSSD in 9 studies [20, 22, 54, 56, 59, 62–64, 68], and PNN50 in 6 studies [22, 57, 62–64, 66]. For frequency domain parameters, the total power was reported in 6 studies [23, 58, 60, 61, 65, 66], LF in 13 studies [20, 21, 23, 25, 54, 56, 57, 59, 62–65, 68], HF in 13 studies [20, 21, 23, 25, 54, 56–59, 62–65] and LF/HF in 12 studies [21, 23, 25, 54, 56–59, 62–65]. We excluded inconsistent data of LF/HF from one study [60].

## 3.7 Meta-analysis on the effect of physical exercise on HRV

After exercise training several time domains indices significantly improved in T2DM patients (Fig 2) i.e. an increased *SDNN* (effect size = 0.59, 95% CI 0.26 to 0.93) [22, 24, 25, 54–57, 59, 62–64, 66, 68], *RMSSD* (0.62, 0.28 to 0.95) [20, 22, 54, 56, 59, 62–64, 68], *PNN50* (0.62, 0.23 to 1.00) [22, 57, 62–64, 66] (S3–S6 Figs). For frequency domains (Fig 3), *LF* decreased (-0.37, -0.69 to -0.05) [20, 21, 23, 25, 54, 56, 57, 59, 62–65, 68], *HF* (0.58, 0.16 to 0.99) [20, 21, 23, 25, 54, 56–59, 62–65], and *LF/HF* (-0.52, -0.79 to -0.24) [20, 21, 23, 25, 54, 56–59, 62–65] increased. All aforementioned meta-analyses had a high degree of heterogeneity (> 60%). *TP* did not differ between groups (0.03, -0.18 to 0.23) [23, 58, 60, 61, 65, 66] (S7–S10 Figs). None of these parameters varied in control groups from RCTs [25, 54, 56–58] (S3–S10 Figs).

## 3.8 Meta-analysis stratified by type of exercise

After **endurance** training, all time and frequency domains measures were improved in T2DM patients: *SDNN* (effect size = 0.65, 95%CI 0.19 to 1.10) [20, 22, 55, 59, 62–64], *RMSSD* (0.66, 0.21 to 1.11) [22, 59, 62–64, 66], *PNN50* (0.87, 0.57 to 1.18) [20, 22, 62–64], and *HF* (0.56, 0.18 to 0.94) [20, 21, 23, 59, 62–65] were significantly higher; *LF* (-0.55, -0.95 to -0.15) [20, 21, 23, 59, 62–65] and *LF/HF* (-0.49, -0.74 to -0.24) [21, 23, 59, 62–65] were significantly lower. After **resistance** training, only *LF* (-0.9, -1.56 to -0.30) [23] and *LF/HF* (-0.96, -1.59 to -0.33) [23] were significantly lower, without any changes for other time (*SDNN*, *RMSSD*, and *PNN50*) and frequency (*HF*) domain parameters. After **combined endurance and resistance** training, there were no changes in any of HRV parameters. After **HIIT**, only two of the aforementioned parameters were improved: *RMSSD* (1.26, 0.58 to 1.94) [56] and *LF/HF* (-1.63, -2.64 to -0.62) [25, 56] (Figs 2 and 3).

For **comparisons between type of exercises**, metaregressions showed that *LF/HF* was more improved after endurance (0.55, 0.11 to 0.98) or resistance (1.022, 0.23 to 1.81) training compared with combined endurance and resistance training, and was more improved after HIIT compared with endurance (-1.19, -1.89 to -0.48) or combined endurance and resistance training (-1.73, -2.49 to -0.98). *PNN50* was more improved after endurance training compared with combined endurance and resistance training (-1.12, -2.07 to -0.161) (Fig 4).

## 3.9 Meta-analysis stratified by type of supervision

After a **supervised** training, *SDNN* (effect size = 0.84; 95%CI 0.42 to 1.25) [22, 24, 54, 54–56, 62–64], *RMSSD* (0.82, 0.43 to 1.21) [22, 54, 56, 62–64], *pNN50* (0.85, 0.55 to 1.15) [20, 22, 62–64] were significantly higher; *LF* (-0.61, -1.01 to -0.21) [23, 54, 56, 62–65] and *LF/HF* (-0.72, -1.07 to -0.37) [23, 54, 56, 62–65] were significantly lower, *HF* (0.82, 0.26 to 1.37) [23, 54, 56, 62–65] was significantly higher. After a **partially supervised** training, there were no changes in any HRV parameters except an increase in *LF* (0.33, 0.01 to 0.65) [21, 57, 58]. After **unsupervised** training, only *LF/HF* (-1.09, -1.99 to -0.19) [25] was significantly lower (Figs 2 and 3).

| | n studies (subgroups) | I-squared (%) | | Effect size (95% CI) | Weight (%) |
|---|---|---|---|---|---|
| **RR** | | | | | |
| **by type of exercise** | | | | | |
| Endurance | 3 (4) | 0.0 | | 0.19 (-0.29 to 0.67) | 45.5 |
| Resistance & Endurance | 1 (1) | - | | 0.43 (-0.17 to 1.02) | 29.1 |
| High intensity interval training | 2 (2) | 0.0 | | 0.22 (-0.42 to 0.86) | 25.4 |
| **by supervised or not** | | | | | |
| Supervised | 2 (3) | 0.0 | | 0.20 (-0.34 to 0.73) | 22.0 |
| Partially supervised | 1 (1) | - | | 0.43 (-0.17 to 1.02) | 25.6 |
| Nonsupervised | 1 (1) | 0.0 | | 0.31 (-0.47 to 1.22) | 52.3 |
| **Overall** | **5 (7)** | **0.0** | | **0.27 (-0.06 to 0.60)** | **100.0** |
| **SDNN** | | | | | |
| **by type of exercise** | | | | | |
| Endurance | 7 (9) | 68.4 | | 0.65 (0.19 to 1.10) | 52.4 |
| Resistance | 1 (1) | - | | 0.41 (-0.25 to 1.07) | 6.5 |
| Resistance & Endurance | 3 (3) | 0.0 | | 0.34 (-0.04 to 0.72) | 18.8 |
| High intensity interval training | 4 (4) | 86.1 | | 0.67 (-0.47 to 1.80) | 22.4 |
| **by supervised or not** | | | | | |
| Supervised | 8 (11) | 71 | | 0.84 (0.42 to 1.25) | 72.6 |
| Partially supervised | 2 (2) | 0.0 | | 0.31 (-0.11 to 0.72) | 15.4 |
| Nonsupervised | 2 (2) | 0.0 | | 0.03 (-0.59 to 0.65) | 12.0 |
| **Overall** | **13 (17)** | **69.5** | | **0.59 (0.26 to 0.93)** | **100.0** |
| **rMSSD** | | | | | |
| **by type of exercise** | | | | | |
| Endurance | 6 (8) | 66.5 | | 0.66 (0.21 to 1.11) | 65.4 |
| Resistance | 1 (1) | - | | 0.20 (-0.45 to 0.86) | 9.3 |
| Resistance & Endurance | 2 (2) | 0.0 | | 0.29 (-0.22 to 0.80) | 16.3 |
| High intensity interval training | 1 (1) | - | | 1.26 (0.78 to 1.94) | 9.01 |
| **by supervised or not** | | | | | |
| Supervised | 6 (8) | 61.6 | | 0.82 (0.43 to 1.21) | 81.0 |
| Partially supervised | 1 (1) | - | | 0.28 (-0.32 to 0.87) | 11.4 |
| Nonsupervised | 1 (1) | - | | 0.24 (-0.69 to 1.17) | 7.6 |
| **Overall** | **9 (12)** | **60.8** | | **0.62 (0.28 to 0.95)** | **100.0** |
| **pNN50** | | | | | |
| **by type of exercise** | | | | | |
| Endurance | 5 (6) | 22 | | 0.87 (0.57 to 1.18) | 73.4 |
| Resistance | 1 (1) | - | | 0.31 (-0.35 to 0.97) | 12.7 |
| Resistance & Endurance | 1 (1) | - | | -0.23 (-0.80 to 0.34) | 13.9 |
| **by supervised or not** | | | | | |
| Supervised | 4 (6) | 26.2 | | 0.85 (0.55 to 1.15) | 76.7 |
| Partially supervised | 1 (1) | - | | -0.23 (-0.80 to 0.34) | 13.9 |
| Nonsupervised | 1 (1) | - | | 0.13 (-0.79 to 1.06) | 9.4 |
| **Overall** | **6 (8)** | **64.9** | | **0.62 (0.23 to 1.01)** | **100.0** |

-1 0 1 2

**Fig 2. Summary of meta-analysis on the effect of exercise training on time domain parameters of HRV in T2DM patients—stratified by type of exercise and type of supervision.**

| | n studies (subgroups) | I-squared (%) | Effect size (95% CI) | Weight (%) |
|---|---|---|---|---|
| **TP** | | | | |
| **by type of exercise** | | | | |
| Endurance | 5 (7) | 0.0 | -0.10 (-0.35 to 0.16) | 64.8 |
| Resistance | 1 (2) | 0.0 | -0.26 (-0.85 to 0.34) | 11.9 |
| Resistance & Endurance | 1 (2) | 0.0 | 0.52 (0.09 to 0.95) | 23.3 |
| **by supervised or not** | | | | |
| Supervised | 2 (6) | 0.0 | -0.23 (-0.54 to 0.07) | 45.6 |
| Partially supervised | 3 (4) | 0.0 | 0.27 (-0.02 to 0.56) | 49.5 |
| Nonsupervised | 1 (1) | - | 0.00 (-0.92 0.92) | 4.9 |
| **Overall** | **6 (11)** | **0.0** | **0.03 (-0.18 to 0.23)** | **100** |
| **LF** | | | | |
| **by type of exercise** | | | | |
| Endurance | 8 (16) | 79.6 | -0.55 (-0.95 to -0.15) | 66.4 |
| Resistance | 1 (2) | 0.0 | -0.93 (-1.56 to -0.3) | 7.7 |
| Resistance & Endurance | 3 (4) | 0.0 | 0.29 (-0.03 to 0.61) | 17.4 |
| High intensity interval training | 2 (2) | 0.0 | 0.32 (-0.18 to 0.82) | 8.5 |
| **by supervised or not** | | | | |
| Supervised | 7 (16) | 81.1 | -0.61 (-1.01 to -0.21) | 73.0 |
| Partially supervised | 3 (4) | 0.0 | 0.33 (0.01 to 0.65) | 18.7 |
| Nonsupervised | 2 (2) | 0.0 | 0.02 (-0.60 to 0.64) | 8.3 |
| **Overall** | **13 (24)** | **78.9** | **-0.37 (-0.69 to -0.05)** | **100** |
| **HF** | | | | |
| **by type of exercise** | | | | |
| Endurance | 8 (16) | 77.3 | 0.56 (0.18 to 0.94) | 72.8 |
| Resistance | 1 (2) | 85.3 | 0.79 (-0.89 to 2.48) | 8.6 |
| Resistance & Endurance | 3 (3) | 0.0 | 0.25 (-0.13 to 0.63) | 14.0 |
| High intensity interval training | 2 (2) | 98.7 | 20.60 (-20.7 to 61.9) | 4.6 |
| **by supervised or not** | | | | |
| Supervised | 7 (16) | 89.0 | 0.82 (0.26 to 1.37) | 75.4 |
| Partially supervised | 3 (3) | 0.0 | 0.33 (-0.06 to 0.71) | 15.1 |
| Nonsupervised | 2 (2) | 0.0 | -0.04 (-0.66 to 0.58) | 9.5 |
| **Overall** | **13 (23)** | **85.6** | **0.58 (0.16 to 0.99)** | **100** |
| **LF/HF** | | | | |
| **by type of exercise** | | | | |
| Endurance | 7 (10) | 18.4 | -0.49 (-0.74 to -0.24) | 55.7 |
| Resistance | 1 (2) | 0.0 | -0.96 (-1.59 to -0.33) | 9.6 |
| Resistance & Endurance | 3 (4) | 0.0 | 0.06 (-0.26 to 0.38) | 24.5 |
| High intensity interval training | 2 (2) | 65.2 | -1.63 (-2.64 to -0.62) | 10.2 |
| **by supervised or not** | | | | |
| Supervised | 7 (11) | 61.9 | -0.72 (-1.07 to -0.37) | 68.3 |
| Partially supervised | 3 (4) | 0.0 | 0.02 (-0.30 to 0.33) | 26.5 |
| Nonsupervised | 1 (1) | - | -1.09 (-1.99 to -0.19) | 5.2 |
| **Overall** | **12 (18)** | **61.1** | **-0.52 (-0.79 to -0.24)** | **100** |

-2    -1    0    1    2

**Fig 3. Summary of meta-analysis on the effect of exercise training on frequency domain parameters of HRV in T2DM patients—stratified by type of exercise and by type of supervision.**

| Variables | n subgroups | I-squared | Coefficient (95% CI) | P-value |
|---|---|---|---|---|
| **SDNN** | | | | |
| **Baseline characteristics** | | | | |
| Total cholesterol, mg.dl-1 | 14 | 49.2 | 0.02 (0.008 to 0.03) | **0.003** |
| **Variations in characteristics after exercise** | | | | |
| Hba1c improvement | 19 | 33.7 | 0.08 (0.045 to 0.116) | **0.000** |
| VO$_2$ peak improvement | 13 | 5.9 | 0.06 (0.033 to 0.093) | **0.001** |
| **rMSSD** | | | | |
| **Baseline characteristics** | | | | |
| Time from diagnosis, years | 11 | 40.4 | 0.09 (0.028 to 0.153) | **0.010** |
| Total cholesterol, mg.dl-1 | 12 | 27.1 | 0.01 (0.000 to 0.084) | **0.044** |
| Diastolic blood pressure, mmHg | 10 | 30.1 | 0.08 (0.015 to 0.156) | **0.022** |
| **Variations in characteristics after exercise** | | | | |
| VO$_2$ peak improvement | 8 | 0.0 | 0.05 (0.013 to 0.087) | **0.017** |
| Systolic blood pressure improvement | 8 | 30.1 | 0.24 (0.047 to 0.425) | **0.023** |
| Diastolic blood pressure improvement | 8 | 35.4 | 0.16 (0.02 to 0.31) | **0.032** |
| **pNN50** | | | | |
| **Baseline characteristics** | | | | |
| Time from diagnosis, years | 7 | 0.0 | 0.06 (0.001 to 0.112) | **0.046** |
| Systolic blood pressure, mmHg | 8 | 44.0 | -0.05 (-0.094 to -0.009) | **0.025** |
| **Type of exercise** | | | | |
| Endurance versus End. and Resist. | 8 | 22.0 | 1.12 (0.161 to 2.071) | **0.030** |
| **Type of supervision** | | | | |
| Supervised versus partially supervised | 8 | 26.2 | 1.08 (0.101 to 2.063) | **0.036** |
| **TP** | | | | |
| **Baseline characteristics** | | | | |
| Beta blockers, % | 12 | 0.0 | 0.14 (0.018 to 0.254) | **0.028** |
| **Type of exercise** | | | | |
| Endurance versus End. and Resist. | 11 | 0.0 | -0.62 (-1.198 to -0.032) | **0.041** |
| **Type of supervision** | | | | |
| Supervised versus partially supervised | 11 | 0.0 | -0.50 (-0.997 to -0.007) | **0.048** |
| **LF** | | | | |
| **Baseline characteristics** | | | | |
| Time from diagnosis, years | 23 | 69.1 | -0.09 (-0.148 to -0.029) | **0.006** |
| Biguanides, % | 11 | 12.3 | 0.04 (0.013 to 0.064) | **0.008** |
| Triglycerides, mg.dl-1 | 25 | 71.1 | -0.01 (-0.021 to -0.006) | **0.039** |
| **Variations in characteristics after exercise** | | | | |
| Body mass index improvement | 17 | 75.0 | 0.13 (0.01 to 0.242) | **0.036** |
| **Type of supervision** | | | | |
| Supervised versus partially supervised | 22 | 76.2 | -1.00 (-1.834 to -0.077) | **0.035** |
| **LF/HF** | | | | |
| **Baseline characteristics** | | | | |
| Biguanides, % | 10 | 0.0 | 0.04 (0.012 to 0.062) | **0.009** |
| Calcium channel blockers, % | 8 | 65.3 | 0.08 (0.004 to 0.154) | **0.042** |
| ACE inhibitors, % | 11 | 56.8 | 0.01 (0 to 0.026) | **0.043** |
| **Type of exercise** | | | | |
| Endurance versus End. and Resist. | 18 | 4.8 | -0.55 (-0.982 to -0.113) | **0.017** |
| Endurance versus HIIT | 18 | 4.8 | 1.19 (0.48 to 1.894) | **0.003** |
| Resistance versus End. and Resist. | 18 | 4.8 | -1.02 (-1.811 to -0.232) | **0.015** |
| End. and Resist. versus HIIT | 18 | 4.8 | 1.73 (0.982 to 2.487) | **0.000** |
| **Variations in characteristics after exercise** | | | | |
| VO$_2$ peak improvement | 11 | 0.0 | -0.05 (-0.081 to -0.019) | **0.005** |
| **Type of supervision** | | | | |
| Supervised versus partially supervised | 16 | 53.7 | -0.70 (-1.378 to -0.02) | **0.045** |

-2   -1   0   1   2

**Fig 4. Metaregressions on factors influencing the effect of exercise training on HRV parameters in T2DM patients.**

For **comparisons between type of supervision**, metaregressions showed that *PNN50* (1.082, 0.10 to 2.06), *LF* (0.96, 0.08 to 1.83) and *LF/HF* (0.70, 0.02 to 1.38) were more improved after supervised exercise training compared to partially supervised (Fig 4).

### 3.10 Others metaregressions

F*requency of exercise* session per week, as well as *session duration* or total *duration of training* were not significantly associated with the variation of any HRV parameters. Patients who improved the most their HRV parameters following exercise training were those with the longest *time from diagnosis of T2DM* (associated with an increased RMSSD and PNN50 following exercise training: 0.09, 0.03 to 0.15, p = 0.010, and 0.06, 0.001 to 0.11, p = 0.046, respectively; and a decrease in LF: -0.09, -0.15 to -0.03, p = 0.006), highest *total cholesterol* levels at baseline (associated with an increase SDNN and RMSSD following exercise training: 0.02, 0.01 to 0.03, p = 0.003, and 0.01, 0.001 to 0.08, p = 0.044, respectively), *triglycerides* (associated with a decrease in LF: -0.01, -0.02 to -0.006, p = 0.039), *diastolic blood pressure* (associated with an increase in RMSSD: 0.08, 0.02 to -0.16, p = 0.022). Patients using biguanides improved less LF and LF/HF after exercise training (0.04, 0.01 to 0.06, p = 0.008 and 0.04, 0.01 to 0.06, p = 0.009, respectively), as well as those using *calcium channel blockers* and *ACE inhibitors* (0.08, 0.004 to 0.15, p = 0.042 and 0.01, 0.001 to 0.03, p = 0.043 both for LF/HF, respectively), whereas users of *beta blockers* had a greater increase in TP (0.14, 0.02 to 0.25, p = 0.028) (Fig 4). Other variables (age, gender, body mass index, smoking, Hba1c, VO$_2$peak, insulin therapy, duration or condition of ECG recording–deep breathing or not) were not significantly associated with the variation of any outcomes.

An improvement in *BMI* after exercise training was significantly associated with an improvement in LF (0.13, 0.01 to 0.24, p = 0.036). Similarly, *improvement of Hba1c* and of SDNN were linked (-0.08, -0.12 to -0.05, p < 0.001), as well as for *V0$_2$ peak* and SDNN (0.06, 0.03 to 0.09, p = 0.001), RMSSD (0.05, 0.01 to 0.09, p = 0.017) and LF/HF (-0.05, -0.08 to -0.02, p = 0.005). I*mprovement of systolic and diastolic blood* pressure were linked to RMSSD improvement (-0.24; -0.43 to -0.05, p = 0.023; and -0.17; -0.31 to -0.02, p = 0.032, respectively) (Fig 4).

### 3.11 Sensitivity analyses

Funnel plots of meta–analyses are presented in S11 Fig. Meta–analyses were reperformed after the exclusion of studies that were not evenly distributed around the base of the funnel and showed similar results. The few studies with the highest level of proof (maximum 6 randomized studies per parameter and maximum 4 RCT per parameter) demonstrated an increase in HF and a decrease in the LF/HF ratio (S12 Fig).

## 4. Discussion

The main findings were that exercise training improved HRV in T2DM patients, with a decrease in sympathetic activity and an increase in parasympathetic activity. Endurance training demonstrated the strongest benefits on HRV parameters. Supervised training improved most HRV parameters, without influence of duration and frequency of training. Patients who benefited the most from exercise training were those with a longer time from diagnosis of T2DM and dyslipidaemia. Improvement in BMI, Hba1c, V0$_2$ peak and blood pressure after exercise training were linked to HRV improvements.

### 4.1 The benefits of exercise training on HRV in T2DM

There are overwhelming evidence that regular physical activity is associated with a reduced risk for all-cause mortality, and several chronic medical conditions [69]. Most international physical activity guidelines recommend to meet the goal of 150 min/week of moderate-to-vigorous intensity physical activity (MVPA) or 75 min/week of vigorous intensity physical activity

[70]. In T2DM patients, exercise leads to better glycemic control, insulin signaling, and blood lipids, reduced low-grade inflammation and improved vascular function [71]. To prevent cardiac autonomic neuropathy, a multifactorial approach is recommended [14], as it can reduce the risk of cardiac autonomic neuropathy progression by 68% [11–13]. Considering that cardiac autonomic neuropathy is a predictor for cardiometabolic events in T2DM patient [72], our meta-analysis showed strong evidence that physical exercise training can improve HRV, both in time and frequency domains. Then, physical exercise training could be a cost-effective intervention to prevent or slow down the cardiac autonomic neuropathy progression in T2DM [73]. Exercise training can improve vagal tone and hence decrease lethal arrhythmias. Even though mechanisms are not yet fully understood, angiotensin II and nitric oxide (NO) are potential mediators of the effects of exercise on vagal tone improvement [74].

## 4.2 Which type of exercise training?

Opinions differ over the exercise modalities that best limit cardiovascular risk [75]. In patients with metabolic syndrome, it has been shown that mixed training with high-intensity increased visceral fat loss, and that training with high-resistance intensity resulted in faster improvement [76]. Greater improvements in sympathovagal balance were demonstrated for patients with metabolic syndrome following a moderate intensity of training; whereas greater decreases in mean 24-hour heart rate were shown for high-intensity resistance training [39]. But the vast majority of the literature about the effects of exercise on glycemic parameters in T2DM has been centered on interventions involving aerobic exercise and there is ample evidence that aerobic exercise is a tried-and-true exercise modality for managing and preventing T2DM [71, 77]. Resistance training showed also benefits for T2DM patients including improvements in glycemic control, insulin resistance, fat mass, blood pressure, strength, and lean body mass [78]. We demonstrated that endurance training led to an improvement of all parameters of time and frequency domain measures, whereas resistance training and HIIT improved only some outcomes. However, the lack of significant results for other modalities than endurance training can be mainly due to insufficient number of studies reporting those modalities of training.

## 4.3 Supervision, frequency, and duration of exercise training

In general, there are strong evidences on the benefits of supervised training [79, 80]. In T2DM, we showed that after supervised exercise training, all HRV parameters were significantly improved whereas none of them was improved after partially supervised exercises and only the LF/HF ratio after unsupervised exercises. Our results are also in accordance with literature in T2DM demonstrating the benefits of supervised training on various parameters such as Hba1c, BMI, blood pressure, dyslipidemia or fitness, in comparison with non-supervised training [15, 81]. Interestingly, those benefits were demonstrated independently of dietary intervention. Despite only including studies assessing an exercise intervention alone, none of the included studies except two [53, 60] followed dietary intake and consequently this might have impacted our results. Furthermore, frequency and volume of exercise have been shown to be linked with metabolic improvement in T2DM [82]. Each aerobic exercise session added within a week may produce an additional reduction of 0.39% in HbA1c level [82]. Nevertheless, we did not find any association between frequency, duration of sessions or duration of interventions and HRV improvements. However, studies did not differ considerably between them (duration of sessions were mainly around 45 minutes, frequency of sessions were mainly around 3 sessions per week, and intensity of exercises were around 60–70% for most studies) without any study assessing the impact of low-intensity training.

## 4.4 Predictors of HRV improvements

Patients who benefited the most from exercise training were those with a longer term diagnosis of T2DM and dyslipidaemia (higher level of total cholesterol and triglycerides), suggesting that benefits might be higher in the most severe patients. Moreover, characteristics of patients such as age or gender may also influence the benefits of exercise training [32, 33, 83, 84]. Some studies reported greater training-induced improvements of HRV in older than in young adults [32, 33], others did not identify any differences [84]. In our meta regressions, age and gender were not associated with any improvement with respect to HRV parameters, BMI or smoking. This suggests that, to some extent, beginning exercise, even late in life can be effective. Finally, it is well known that hypertension is linked with poor HRV [85]. We demonstrated that benefits of exercise training were lower in T2DM patients taking antihypertensive medications, even if literature showed significant improvement in HRV parameters after exercise training in hypertensives patients [86, 87]. T2DM patients using metformin improved less their HRV after exercise compared with T2M patients that did not use metformin. This result may seem contradictory, as metformin has been shown to improve HRV [88]. We also note that most studies (14/21) did not report the use of metformin. Considering that metformin is the first medication to treat T2DM, our results may suffer from a bias of reporting.

## 4.5 Clinical and biological improvements associated with benefits of exercise on HRV

There is strong evidence showing that physical activity is associated with a reduction in all-cause mortality [89]. Exercise training is known to improve several metabolic parameters in T2DM patients such as HbA1c%, serum insulin and glucose, $VO_2peak$ [28, 90] but physiology of exercise benefits on HRV remains unclear. For example, it remains unclear to what extent changes in blood lipids contribute to the cardiovascular benefits of exercise [16, 91]. In our meta-analysis, an improvement in HDL, LDL, total cholesterol or triglycerides serum levels after exercise was not linked with any improvement in HRV parameters, suggesting that improvement in lipid levels would not be associated with exercise benefits [16, 91]. Conversely, we showed association between HRV improvements and improvement of Hba1c, BMI, and $VO_2peak$ after exercise suggesting that these parameters could be key contributors of exercise benefits on HRV [80]. Increased HRV were therefore linked with a better control of T2DM, and with fitness improvement. Beta blockers are known to affect HRV [92]. We cannot conclude that beta blockers influenced response in HRV to exercise, as only one study reported its use [58] and beta blockers being explicitly an exclusion criteria in most studies [19, 21, 23, 25, 52, 55, 56, 59–61, 63, 64].

## 4.6 Limitations

We inherited the limitations of all meta-analyses [93] and the limitations and biases of the individual studies investigated. Furthermore, we conducted the meta-analyses on only published articles, so they are theoretically exposed to publication bias. While the meta-analysis is based on a moderate number of studies, the use of broader keywords in the search strategy limits the number of missing studies. In addition, some studies were monocentric, limiting the generalizability of our results. Moreover, the generalizability of our results (improvement of HRV following exercise training) may also be limited to patients who have a T2DM rather well controlled (as they have an Hba1c 7.5%). Data collections and inclusion/exclusion criteria were not identical within each study, which may have affected our results, as well as heterogeneity due to different study designs. To reduce bias of measures, when a study reported HRV

in different positions [94], we limited data to decubitus measures, as position and conditions of measure may influence HRV. Most studies included were not RCT, precluding robust conclusions for our meta-analyses. For LF and HF, some studies reported measures in both ms$^2$ and normalized units. These reported measures have been included in our meta-analysis and therefore could affecting the weighting of studies. However, we conducted sensitivity analyses with only one or the other unit to verify that it did not affect the results. We also limited the influence of extreme results and heterogeneity by repeating analyses after the exclusion of studies with results not evenly distributed around the funnel plots. Finally, even though in some studies [23, 24, 53, 55, 60, 62–65] patients were asked not to change their dietary intake, an assessment of their dietary intake should have been conducted to verify that the exercise intervention did not modify their eating habits (that could be a confounding factor).

## 5. Conclusion

Exercise training improved HRV parameters in T2DM patients, which may reflect an improvement in the activity of the autonomic nervous system. The level of proof was highest for endurance training (aerobic), whereas resistance (anaerobic) and high-intensity-interval training (alternating short intense anaerobic and less intense exercises) were promising. Supervised training seemed beneficial, whereas insufficient data precluded robust conclusions for duration and frequency of sessions. HRV improvements may be mediated by the improvement in clinical and biological parameters consecutive to exercise training.

## Supporting information

**S1 Checklist. PRISMA checklist.**
(DOCX)

**S1 Appendix. Quality of studies–Scottish Intercollegiate Guidelines Network (SIGN) grids.**
(DOCX)

**S2 Appendix. Quality of studies–Physiotherapy Evidence Database PEDro.**
(PDF)

**S1 Fig. Methodological quality of included articles using Scottish Intercollegiate Guidelines Network (SIGN) scale.** For each item, criteria fulfilled: No: -, Yes: +, Unclear:?, Not applicable: NA.
(TIF)

**S2 Fig. Methodological quality of included articles using PEDro.**
(TIF)

**S3 Fig. Effect of exercise training on RR in T2DM patients.**
(PDF)

**S4 Fig. Effect of exercise training on SDNN in T2DM patients.**
(PDF)

**S5 Fig. Effect of exercise training on RMSSD in T2DM patients.**
(PDF)

**S6 Fig. Effect of exercise training on pNN50 in T2DM patients.**
(PDF)

**S7 Fig. Effect of exercise training on TP in T2DM patients.**
(PDF)

**S8 Fig. Effect of exercise training on LF in T2DM patients.**
(PDF)

**S9 Fig. Effect of exercise training on HF in T2DM patients.**
(PNG)

**S10 Fig. Effect of exercise training on LF/HF in T2DM patients.**
(PDF)

**S11 Fig. Funnel plots.**
(TIF)

**S12 Fig. Summary of meta-analysis on the effect of exercise training on HRV in T2DM patients, using only the studies with the best methodological design (randomized studies, and randomized controlled studies).**
(TIF)

## Acknowledgments

We thank Nathalie Piñol-Domenech, librarian of the Clermont Auvergne University, and the librarians of University Health Sciences Library of Paris for their support in the elaboration of search strategy in databases and collection of full texts of articles.

## Author Contributions

**Conceptualization:** Mathilde Picard, Igor Tauveron, Salwan Magdasy, Thomas Benichou, Frédéric Dutheil.

**Data curation:** Mathilde Picard, Frédéric Dutheil.

**Formal analysis:** Mathilde Picard, Salwan Magdasy, Thomas Benichou, Frédéric Dutheil.

**Investigation:** Mathilde Picard, Salwan Magdasy, Thomas Benichou, Frédéric Dutheil.

**Methodology:** Mathilde Picard, Igor Tauveron, Salwan Magdasy, Thomas Benichou, Reza Bagheri, Valentin Navel, Frédéric Dutheil.

**Project administration:** Igor Tauveron, Salwan Magdasy, Frédéric Dutheil.

**Resources:** Igor Tauveron, Frédéric Dutheil.

**Software:** Mathilde Picard, Frédéric Dutheil.

**Supervision:** Igor Tauveron, Thomas Benichou, Frédéric Dutheil.

**Validation:** Mathilde Picard, Igor Tauveron, Salwan Magdasy, Thomas Benichou, Reza Bagheri, Ukadike C. Ugbolue, Frédéric Dutheil.

**Visualization:** Mathilde Picard, Ukadike C. Ugbolue, Frédéric Dutheil.

**Writing – original draft:** Mathilde Picard, Valentin Navel, Frédéric Dutheil.

**Writing – review & editing:** Mathilde Picard, Reza Bagheri, Ukadike C. Ugbolue, Valentin Navel, Frédéric Dutheil.

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
