## [Decision Letter · Decision Letter 0]

12 Jan 2021

PONE-D-20-40085

Effect of exercise training on heart rate variability in type 2 diabetes mellitus patients: A systematic review and meta-analysis

PLOS ONE

Dear Dr. Navel,

Thank you for submitting your manuscript to PLOS ONE. After careful consideration, we feel that it has merit but does not fully meet PLOS ONE’s publication criteria as it currently stands. Therefore, we invite you to submit a revised version of the manuscript that addresses the points raised during the review process.

We look forward to receiving your revised manuscript.

Kind regards,

Walid Kamal Abdelbasset, Ph.D.

Academic Editor

PLOS ONE

Journal Requirements:

2. Please confirm that you have included all items recommended in the PRISMA checklist including the full electronic search strategy used to identify studies with all search terms and limits for at least one database.

3. Please ensure you have provided details of reasons for study exclusions in the PRISMA flowchart and number of studies excluded for each reason.

4. Please note that PLOS does not permit references to “data not shown.” Authors should provide the relevant data within the manuscript, the Supporting Information files, or in a public repository. If the data are not a core part of the research study being presented, we ask that authors remove any references to these data.

"NO "

7. We note that this manuscript is a systematic review or meta-analysis; our author guidelines therefore require that you use PRISMA guidance to help improve reporting quality of this type of study. Please upload copies of the completed PRISMA checklist as Supporting Information with a file name “PRISMA checklist”.

Reviewers' comments:

Reviewer's Responses to Questions

**Comments to the Author**

1. Is the manuscript technically sound, and do the data support the conclusions?

Reviewer #1: Yes

Reviewer #2: Yes

Reviewer #3: Yes

2. Has the statistical analysis been performed appropriately and rigorously? 

Reviewer #1: Yes

Reviewer #2: Yes

Reviewer #3: Yes

3. Have the authors made all data underlying the findings in their manuscript fully available?

Reviewer #1: Yes

Reviewer #2: Yes

Reviewer #3: Yes

4. Is the manuscript presented in an intelligible fashion and written in standard English?

Reviewer #1: Yes

Reviewer #2: Yes

Reviewer #3: Yes

5. Review Comments to the Author

Reviewer #1: Reviewer comments:

Thank you for giving the opportunity to review this article.

Please edit the entire manuscript for English grammar and syntax for readability.

Abstract

1. Conclusion of the review should be more self-explanatory.

2. Include its clinical significance on physicians, patients and researchers.

Introduction

1. The introduction part is too short and didn’t mention about important key points.

2. Mention in detail about the relation between cardiac autonomic neuropathy, type of exercise and HRV.

3. Define the clinical significance of this study in related to researchers, clinicians and patients.

Methods

4. The data bases searched are much limited in this review.

5. The selection criteria should be more specific (inclusion and exclusion).

6. Mention the kappa score of data extracted reviewers.

7. How come including both randomized and non-randomized controlled trial will give quality reports?

Results

8. Include the risk of bias analysis.

9. Mention the data are analyzed by fixed or random analysis method.

Discussion

10. Explain in detail and its mechanism, how the outcome variables are helpful to change this condition.

11. Conclusion – how come the authors came to the conclusion of changing of sympathetic and parasympathetic activity in T2DM patients?

12. Refine the conclusion according to the objective of your study.

13. Follow author guidelines for the tables.

Reviewer #2: ABSTRACT you should add more details that make your review more clear (types , quality of selected articles, of )

Provide us with Systematic review registration number if you registered in (https://www.crd.york.ac.uk/PROSPERO/#searchadvanced) Web site

I think you did not register (let me know the reason )

In abstract study eligibility criteria (the time range of article included in your review )

in inclusion criteria (you should add the types of studies selected and divided according to types of exercise ,design ,or something like it will make more accuracy of your results )

quality of study selection measurement (it should be added ect ,PEDRO )

your study used score of sign (why ?)

time of selected studies (what’s the optimal timing of your research )

(page no 2 )in abstract (Method: PubMed, Cochrane, Embase, and ScienceDirect databases were searched for studies reporting HRV parameters in T2DM patients before and after exercise training, until September 20th 2020)

(page no 8) in 3.2 Study designs and objectives (Included studies were published from 2003 to 2019 and conducted in various geographic locations)

2.4 Statistical considerations (page 6 an 7)

In this section it should rewritten again to be more clear

3 Results

An initial search produced 6641 possible articles

The end 21 article

Did you use any soft ware when you select these 21 articles ?

Results of individual studies, for all outcomes considered (benefits or harms)

3.3 Inclusion and exclusion criteria of included studies

This title should be (Study characteristics)

This part need rewrite to be more clear

The question is

All selected studies had measured these all subtitle (Aerobic capacity, Blood lipid levels, Blood pressure, BMI, Metabolic control)

If yes

Please rearrange this paragraph and make it in table or add the names of authors in the paragraph

If not (give reasons)

Reviewer #3: Good piece of work and very good effort from the authors in collecting the data. However, there are some certain points need to be corrected or explained before considering this manuscript for publication.

Introduction:

There should be a paragraph explaining the HRV giving what time domain include and what frequency domain include.

Also there are many methods to assess the cardiac autonomic function, why did you specify it only to HRV?

The authors mentioned that studies in HRV in T2DM are scarce, give an explanation and an example to those studies.

Study designs and objectives:

the 21 included studies. please explain which one of them used frequency domain or time domain

Inclusion and exclusion criteria section: it was stated "In most studies, sedentary behavior or low level of physical activity was necessary" why it was necessary, please give full explanation

Metabolic control section: "the mean HbA1c in T2DM patients following exercise training was 7.5 %" The percentage here after 7.5 has no meaning.

Aerobic capacity section: Please identify if VO2 max was measured or Vo2 peak, as in some area of the manuscript Vo2 peak was mentioned. Considering the age of the participants, I doubt that Vo2 max was measured, but please check.

Duration of measures: The validity of HRV less than 2 min is questioned, therefor, I am concerned how recording of HRV of 1 min was included. Look for the task force of the European Society of Cardiology and the North

American Society of Pacing and Electrophysiology. The recomendation for short recording are only valid and specified for frequency domain analysis

Other metaregressions section:

Please report the p value as well as the unit of measurement whether it is ms or normailzed unit.

Also I suggest adding creating a table where the meaning of LF and HF can be easily tracked. For example does LF measures purely sympathetic.

Also when improvement in LF is mentioned, does that indicate a decrease or increase?

In the first paragraph of the discussion, define what is HRV improvement?

Which parameter in HRV represent the improvement in sympathovagal balance

"We also showed that exercise improved less HRV in T2DM patients reporting the use of metformin" This statement is not clear. please clarify.

Also I suggest instead of using decreased HRV, poor HRV.

"Few effects on lipid level profile have been

demonstrated and it remains unclear to what extent changes in blood lipids contribute to the

cardiovascular benefits of exercise" This statement also need clarification.

Overall, there should be pragragraph indicating the effect of different positions while measuring HRV.

Also The effect of different medication especially beta blockers on HRV.

In the conclusion: it is mentioned endurance, and high intensity interval, please clarify what is the difference?

6. PLOS authors have the option to publish the peer review history of their article (what does this mean?). If published, this will include your full peer review and any attached files.

Reviewer #1: No

Reviewer #2: No

Reviewer #3: **Yes: **Ahmad Osailan

---

## [Author Response · Author response to Decision Letter 0]

2 Apr 2021

Walid Kamal Abdelbasset

Academic Editor

PLOS ONE

Dear Editor,

My coauthors and I welcomed the review of our Manuscript PONE-D-20-40085 entitled “Effect of exercise training on heart rate variability in type 2 diabetes mellitus patients: A systematic review and meta-analysis”. We have addressed the comments of the reviewers in a revised manuscript and enclose a point-by-point response.

We would like to thank the reviewers for their insightful comments on the letter which have enabled us to make improvements to the manuscript. Our revision has taken into account all reviewer suggestions and comments and our detailed responses are provided below.

Journal Requirements

[REPLY] We followed the PLOS ONE's style requirements.

2. Please confirm that you have included all items recommended in the PRISMA checklist including the full electronic search strategy used to identify studies with all search terms and limits for at least one database.

[REPLY] Thank you for your relevant comment. We followed the PRISMA Checklist. 

3. Please ensure you have provided details of reasons for study exclusions in the PRISMA flowchart and number of studies excluded for each reason.

[REPLY] Thank you for your relevant comment. We added the number of studies excluded for each reason within the flowchart.

4. Please note that PLOS does not permit references to “data not shown.” Authors should provide the relevant data within the manuscript, the Supporting Information files, or in a public repository. If the data are not a core part of the research study being presented, we ask that authors remove any references to these data.

[REPLY] Thank you for your comment. We added some supplementary materials when needed.

5. Thank you for stating the following financial disclosure: "NO"

At this time, please address the following queries: Please clarify the sources of funding (financial or material support) for your study. List the grants or organizations that supported your study, including funding received from your institution. State what role the funders took in the study. If the funders had no role in your study, please state: “The funders had no role in study design, data collection and analysis, decision to publish, or preparation of the manuscript.” If any authors received a salary from any of your funders, please state which authors and which funders. If you did not receive any funding for this study, please state: “The authors received no specific funding for this work.” Please include your amended statements within your cover letter; we will change the online submission form on your behalf.

[REPLY] Thank you for your comment. We added the following information after the conclusions and before the references: “Funding: The authors received no specific funding for this work”.

[REPLY] Thank you for your comment. Ethical statement is not applicable because of the nature of the article (systematic review of the literature).

7. We note that this manuscript is a systematic review or meta-analysis; our author guidelines therefore require that you use PRISMA guidance to help improve reporting quality of this type of study. Please upload copies of the completed PRISMA checklist as Supporting Information with a file name “PRISMA checklist”.

[REPLY] Thank you for your comment. We uploaded the PRISMA checklist as a supplementary file.

Reviewers' comments

1. Is the manuscript technically sound, and do the data support the conclusions?

Reviewer #1: Yes

Reviewer #2: Yes

Reviewer #3: Yes

[REPLY] Thank you for your positive opinion.

2. Has the statistical analysis been performed appropriately and rigorously?

Reviewer #1: Yes

Reviewer #2: Yes

Reviewer #3: Yes

[REPLY] Thank you for your positive opinion.

3. Have the authors made all data underlying the findings in their manuscript fully available?

Reviewer #1: Yes

Reviewer #2: Yes

Reviewer #3: Yes

[REPLY] Thank you for your positive opinion.

4. Is the manuscript presented in an intelligible fashion and written in standard English?

Reviewer #1: Yes

Reviewer #2: Yes

Reviewer #3: Yes

[REPLY] Thank you for your positive opinion.

5. Review Comments to the Author

Reviewer 1

Thank you for giving the opportunity to review this article.

[REPLY] Thank you for your positive opinion.

Please edit the entire manuscript for English grammar and syntax for readability.

[REPLY] Thank you for comment.

Abstract

1. Conclusion of the review should be more self-explanatory. 2. Include its clinical significance on physicians, patients and researchers.

[REPLY] Thank you for comment. The conclusion of the abstract now reads: “Exercise training improved HRV parameters in T2DM patients which may reflect an improvement in the activity of the autonomic nervous system. The level of proof is the highest for endurance training. Supervised training seemed beneficial.” As reviewer 3 also asked for more details within the methods section of the abstract, we are now far above the limit of 300 words for the abstract. We would be happy to prioritise some points on request.

Introduction

1. The introduction part is too short and didn’t mention about important key points.

2. Mention in detail about the relation between cardiac autonomic neuropathy, type of exercise and HRV.

3. Define the clinical significance of this study in related to researchers, clinicians and patients.

[REPLY] Thank you for comment. The length of the introduction doubled. In particular, we gave further details on the most important key points, the relation between cardiac autonomic neuropathy, type of exercise and HRV, and the clinical significance of this study for researchers, clinicians and patients.

Methods

4. The data bases searched are much limited in this review.

[REPLY] Thank you for comment. The data bases searched were PubMed, Cochrane Library, ScienceDirect and Embase. Most meta-analyses include the data bases that we have used or most often less, including meta-analysis published in high-ranking top journals (Medline and Web of Science for Chersich MF et al. BMJ. 2020 Nov 4;371:m3811. doi: 10.1136/bmj.m3811; Medline, Embase, and Cochrane CENTRAL for Palmer et al. BMJ. 2021 Jan 13;372:m4573. doi: 10.1136/bmj.m4573; Medline, Embase, and Cochrane CENTRAL In Epure AM et al. PLoS Med. 2020 Nov 23;17(11):e1003414. doi: 10.1371/journal.pmed.1003414; PubMed, Embase, Cochrane Library, and Web of Science Eur Heart J. 2020 Dec 7; 41(46): 4415–442; doi: 10.1093/eurheartj/ehaa793). We have chosen a combination of several databases (PubMed, Cochrane Library, ScienceDirect and Embase) that are complementary: a data base indexing the best articles (PubMed), a database indexing reviews to enlarge our search (Cochrane) and two very large databases indexing a large number of articles (ScienceDirect and Embase), but with a lot of noise. In total, several dozen million articles are indexed in those databases.

5. The selection criteria should be more specific (inclusion and exclusion).

[REPLY] Thank you for comment. We added some details (i.e. the fact that articles needed to report HRV data both at baseline and after exercise training in a T2D group, and that a control group without exercise was not needed). The selection criteria section now reads: “We reviewed all studies reporting the effect of exercise training on HRV in T2DM patients. Animals studies were excluded. The PubMed, Cochrane Library, Science Direct and Embase databases were searched until September 20th 2020, with the following keywords: diabetes AND (exercise OR physical) AND (“heart rate variability” OR HRV). The search was not limited to specific years and no language restrictions were applied. To be included, studies needed to describe our primary outcome variables i.e. HRV data before and after exercise training in T2DM patients, with or without a control group (no physical activity intervention). We excluded studies that assessed the effects of other intervention (such as dietary or psychological intervention) in combination with exercise training. Conferences, congress or seminars, were excluded. In addition, reference lists from all publications meeting the inclusion criteria were manually searched to identify any further studies not found through the electronic search. Ancestry searches were also completed on previous reviews to locate other potentially eligible primary studies.”

6. Mention the kappa score of data extracted reviewers.

[REPLY] Thank you for comment. We added the kappa score between reviewers at the beginning of the Results section. The second sentence of the Results section now reads: “Removal of duplicates and use of the selection criteria reduced the number of articles reporting the effect of exercise on HRV in T2DM patients to 21 articles (inter-reader agreement κ = 0.89)”.

7. How come including both randomized and non-randomized controlled trial will give quality reports?

[REPLY] Thank you for comment. Because of the heterogeneity of study designs and because of the low number of RCT, we did not show sensitivity analyses considering only studies with the highest level of proof. However, we agree that, despite meta-analyses on few studies cannot warrant evidence-based conclusions, it still gives an information to readers. Therefore, we added the following sentence within the Methods – statistics section “Lastly, we reperformed the aforementioned meta-analyses only using the studies with the highest level of proof, i.e. only on randomised studies and only on randomized controlled studies.” The Results – sensitivity analyses section reads: “The few studies with the highest level of proof (maximum 6 randomized studies per parameter and maximum 4 RCT per parameter) demonstrated an increase in HF and a decrease in the LF/HF ratio (S12 Fig).” The Limitations section reads: “Most studies included were not RCT, precluding robust conclusions for our meta-analyses.” We also added a supplementary figure synthesizing the aforementioned sensitivity meta-analyses: “S12 Fig. Summary of meta-analysis on the effect of exercise training on HRV in T2DM patients, using only the studies with the best methodological design (randomized studies, and randomized controlled studies)”.

Results

8. Include the risk of bias analysis.

[REPLY] Thank you for comment. The risk of bias analysis is in section 3.1. Quality of articles. We have expanded this section with the additional use of PEDRO. The Methods section now reads: “In addition, we also used the Physiotherapy Evidence Database (PEDro) scale for a complementary overview of studies quality, with score ranging from 0 to 10 – 10 being the highest score (S3 Fig).” The Results section now reads: “Using the PEDro scale, score ranged from 2 [19,20,24] to 6 [23,54–56] out of 10 (S3 Appendix and S2 Fig)”.

9. Mention the data are analyzed by fixed or random analysis method.

[REPLY] Thank you for comment. The “Statistical considerations” section reads: “We conducted random effects meta–analyses (DerSimonian and Laird approach) when data could be pooled [37].” Reference 37 is the following: “DerSimonian R, Laird N. Meta-analysis in clinical trials. Control Clin Trials. 1986;7: 177–188.”

Discussion

10. Explain in detail and its mechanism, how the outcome variables are helpful to change this condition.

[REPLY] Thank you for comment. We added the following sentences within the Discussion: “Exercise training can improve vagal tone and hence decrease lethal arrhythmias. Even if mechanisms are not yet fully understood, angiotensin II and nitric oxide (NO) are potential mediators of the effects of exercise on vagal tone improvement (REF).” The following reference was added: “Routledge FS, Campbell TS, McFetridge-Durdle JA, Bacon SL. Improvements in heart rate variability with exercise therapy. Can J Cardiol. Jun-Jul 2010;26(6):303-12. doi: 10.1016/s0828-282x(10)70395-0”.

11. Conclusion – how come the authors came to the conclusion of changing of sympathetic and parasympathetic activity in T2DM patients?

[REPLY] Thank you for comment. We agree that there are ongoing debate on the signification of some HRV parameters in relation with sympathetic and parasympathetic activity. Therefore we proposed a more generalisable comment. The conclusion now reads: “Exercise training improved HRV parameters in T2DM patients, with may reflect an improvement in the activity of the autonomic nervous system.”

12. Refine the conclusion according to the objective of your study.

[REPLY] Thank you for comment. We totally agree that there were three objectives of our study and that our conclusion only summarized the first two objectives. The objectives of our study were to study: “1) the impact of exercise on HRV in patients with T2DM, 2) depending on modalities of exercise such as the type of exercise, its supervision or not, or duration and frequency of sessions, 3) and depending on characteristics of patients.” Consequently, our conclusion has been refined according to the objective of our study. Our conclusion now reads: “Exercise training improved HRV parameters in T2DM patients, with may reflect an improvement in the activity of the autonomic nervous system. The level of proof is the highest for endurance training, despite resistance and high-intensity-interval training may also be promising. Supervised training seemed beneficial, whereas insufficient data precluded robust conclusions for duration and frequency of sessions. HRV improvements may be mediated by the improvement in clinical and biological parameters consecutive to exercise training.”

13. Follow author guidelines for the tables.

[REPLY] Thank you for comment. The Table 1 has been updated. In particular, we removed the use of returns to align content across rows and columns. Each content is now in separate cells.

Reviewer 2

ABSTRACT

You should add more details that make your review more clear (types, quality of selected articles, of)

[REPLY] Thank you for comment. We added all the suggested information (types of studies, more study eligibility criteria and the time range of article included, as well as sensitivity analyses based on quality of selected articles

Provide us with Systematic review registration number if you registered in (https://www.crd.york.ac.uk/PROSPERO/#searchadvanced) Web site. I think you did not register (let me know the reason)

[REPLY] Thank you for comment. At the time of registration on PROSPERO, we were advised that because of the COVID-19 pandemics and the high number of studies submitted, the NIHR would not process our registration before several months – as they decided to prioritise studies with authors from UK (because PROSPERO is funded by the National Institute for Health Research (NIHR) from UK). Secondly, even if we agree that PROSPERO registration may guarantee for better quality of systematic review and meta-analyses, for some reasons, high ranked journal such as Lancet, JAMA, Sports Med, did not specifically require a PROSPERO registration. See for example “Population-level impact and herd effects following the introduction of human papillomavirus vaccination programmes: updated systematic review and meta-analysis. Lancet. 2019 Aug 10;394(10197):497-509. doi: 10.1016/S0140-6736(19)30298-3”; “Association of general anesthesia vs procedural sedation with functional outcome among patients with acute ischemic stroke undergoing thrombectomy: a systematic review and meta-analysis. JAMA. 2019 Oct 1;322(13):1283-1293. doi: 10.1001/jama.2019.11455; “Active commuting and multiple health outcomes: a systematic review and meta-analysis. Sports Med. 2019 Mar;49(3):437-452. doi: 10.1007/s40279-018-1023-0.” Plos One also most often publish meta-analyses without a PROSPERO registration, see for example the following articles: “Meta-analysis and sustainability of feeding slow-release urea in dairy production, doi: 10.1371/journal.pone.0246922”; “Meta-analysis of the correlation between serum uric acid level and carotid intima-media thickness, doi: 10.1371/journal.pone.0246416”; “Efficacy and safety of diazoxide for treating hyperinsulinemic hypoglycemia: A systematic review and meta-analysis, doi: 10.1371/journal.pone.0246463”.

In abstract study eligibility criteria (the time range of article included in your review)

In inclusion criteria (you should add the types of studies selected and divided according to types of exercise, design, or something like it will make more accuracy of your results)

quality of study selection measurement (it should be added etc., PEDRO)

time of selected studies (what’s the optimal timing of your research)

(page no 2) in abstract (Method: PubMed, Cochrane, Embase, and ScienceDirect databases were searched for studies reporting HRV parameters in T2DM patients before and after exercise training, until September 20th 2020)

[REPLY] Thank you for comment. We added all the suggested information (types of studies, more study eligibility criteria and the time range of article included, as well as sensitivity analyses based on quality of selected articles). The sentences now reads: “PubMed, Cochrane, Embase, and ScienceDirect databases were searched for all studies reporting HRV parameters in T2DM patients before and after exercise training, until September 20th 2020, without limitation to specific years. We conducted random-effects meta-analysis depending on type of exercise for each HRV parameters: RR–intervals (or Normal to Normal intervals – NN), standard deviation of RR intervals (SDNN), percentage of adjacent NN intervals varying by more than 50 milliseconds (pNN50), root mean square of successive RR-intervals differences (RMSSD), total power, Low Frequency (LF), High Frequency (HF) and LF/HF ratio. Sensitivity analyses were computed on studies with the highest quality. Results: We included 21 studies (9 were randomized) for a total of 523 T2DM patients”. Please note that we are now far above the limit of 300 words for the abstract. We would be happy to prioritise some points on request.

your study used score of SIGN (why ?)

[REPLY] Thank you for comment. Unfortunately, there are still no guidelines and clear recommendation about the choice of checklists in the evaluation of methodological quality of studies included in a meta-analysis. For example, even within the same Journal, a wide range of checklist can be used (see for example the following articles that all used different checklists: “Lancet 2019;394(10197):497-509, doi: 10.1016/S0140-6736(19)30298-3”; “JAMA 2019;322(13):1283-1293, doi: 10.1001/jama.2019.11455”; “Sports Med 2019;49(3):437-452. doi: 10.1007/s40279-018-1023-0”; “Plos One, doi: 10.1371/journal.pone.0246922”; “Plos One, doi: 10.1371/journal.pone.0246416”; “Plos One, doi: 10.1371/journal.pone.0246463”. We used the SIGN score as there are several SIGN checklist: one per study design. Therefore, as we included several type of study designs within included articles, the SIGN checklists permitted to keep consistency in the evaluation of the quality of studies. SIGN is also widely accepted and is often used in meta-analysis. See for example hundreds meta-analyses using SIGN checklist: “https://pubmed.ncbi.nlm.nih.gov/?term=meta-analysis+%22SIGN%22&sort=date”. However, as suggested, we also used PEDRO in addition to SIGN to give readers a complementary overview of studies quality. The evaluation of studies quality using the PEDRO checklist is now available in Electronic Supplementary Material Appendix 3.

METHODS

(page no 8) in 3.2 Study designs and objectives (Included studies were published from 2003 to 2019 and conducted in various geographic locations)

[REPLY] Thank you for comment. The methods section reads: “The search was not limited to specific years and no language restrictions were applied.” The result is that we retrieved studies published from 2003 to 2019.

2.4 Statistical considerations (page 6 and 7)

In this section it should rewritten again to be more clear

[REPLY] Thank you for comment. We rewrote the statistical considerations section. It now reads: “We conducted meta–analyses on the effect of exercise on HRV parameters in T2DM. P values less than 0.05 were considered statistically significant. For the statistical analysis, we used Stata software (version 16, StataCorp, College Station, US) [44–48]. Main characteristics were synthetized for each study population and reported as mean ± standard-deviation (SD) for continuous variables and number (%) for categorical variables. We conducted random effects meta–analyses (DerSimonian and Laird approach) when data could be pooled (more than five data for the same outcome) [49]. A particular attention was paid for short recording (1 minute) of HRV parameters [50]. First, we calculated the effect size (ES, standardised mean differences – SMD) [51] of each HRV parameter after exercise compared to baseline (before exercise) in T2DM. ES is a unitless measure centered at zero if HRV did not differ between measures before and after exercise. A positive ES denoted higher levels of the tested HRV parameter in T2DM patients after exercise. An ES of 0.8 reflects a large effect, 0.5 a moderate effect, and 0.2 a small effect. As ES is a unitless measure and as we compared data after and before exercise, frequency-domain HRV parameters measured in ms2 or in normalized unit (nu) were combined. Then, we conducted meta-analyses stratified on type of exercise (endurance, resistance, mixed, HIIT), supervision of exercise or not. We also conducted meta-analysis on controls to verify the absence of changes within each HRV parameter. We searched for potential publication bias using funnel plots of all aforementioned meta–analyses and we evaluated heterogeneity by examining forest plots, confidence intervals (CI) and I-squared (I²). A low heterogeneity is reflected by I² values <25%, modest for 25–50%, and high for >50%. We verified the strength of our results by conducting further meta–analyses (sensitivity analyses) after exclusion of studies that were not evenly distributed around the base of the funnel. Lastly, we reperformed the aforementioned meta-analyses only using the studies with the highest level of proof, i.e. only on randomised studies and only on randomized controlled studies. When possible (sufficient sample size), meta–regressions were proposed to study the relationship between changes in HRV parameter (RR intervals, RMSSD, pNN50, SDNN, total power, LF, HF, LF/HF) and clinically relevant parameters such as characteristics of intervention (type of exercise, supervised or not, duration and number of sessions, frequency, intensity), clinical parameters (time from T2DM diagnosis, HbA1c, etc.), sociodemographic (age, sex, etc.), or methods of measurement of HRV, and their changes when pertinent. Results were expressed as regression coefficients and 95% CI.”

RESULTS

An initial search produced 6641 possible articles. The end 21 articles. Did you use any soft ware when you select these 21 articles ?

[REPLY] Thank you for comment. The sentence now reads: “After removal of duplicates using Zotero® software, all possible articles were manually checked by two authors. The use of the selection criteria reduced the number of articles reporting the effect of exercise on HRV in T2DM patients to 21 articles (inter-reader agreement κ = 0.89) (Fig 1).” 

Results of individual studies, for all outcomes considered (benefits or harms)

[REPLY] Thank you for comment. The Table 1 now includes significant increase (�) or decrease (�) for each HRV parameter (differences between groups or after vs before exercise training). We also added a column “other outcomes” in Table 1 summarizing all other outcomes mentioned within each individual included study.

3.3 Inclusion and exclusion criteria of included studies. This title should be (Study characteristics). This part need rewrite to be more clear

[REPLY] Thank you for comment. The section now reads: “Study characteristics: inclusion and exclusion criteria. Included patients had to have T2DM, without further details for most studies. Inclusion criteria for TD2M patients were biological in five studies (fasting glucose > 126 mg/dl [19,22,52,60,61] or glucose levels > 200 mg/dl after an oral glucose tolerance test [60,61]). Most studies included patients with sedentary behavior or low level of physical activity [21–23,56,60–62,65,66], i.e. less than 3 hours of physical activity per week [65] or less than 60 min moderate vigorous activity per week [25], or not being involved in regular physical activity [20–23,60–62]. Some studies required patients to be aged over 35 [53], 40 [23,58,60,61] or 65 [59] years old, or under 60 [53,60,61], 65 [55], 70 [23], 74 [59] years old, or between 12 and 19 years old [20]. Nine studies included sex-specific populations: men [22,53,56,57,63,64] or women [21,54,59]. Six studies included patients according to their body mass index (BMI): >30 [21,56,60,61,64] or between 24 and 36 kg.m-2 [23]. The main exclusion criteria were: smoking [21,23,24,55,56,59–61,63–65], exogenous insulin [21,22,25,55,56,59,63], beta blockers or arrythmia [19,21,23,25,52,55,56,59–61,63,64], and cardiovascular disease [21,23–25,52,53,55,56,58,60,61,63–66].”

The question is:

All selected studies had measured these all subtitle (Aerobic capacity, Blood lipid levels, Blood pressure, BMI, Metabolic control)

If yes: Please rearrange this paragraph and make it in table or add the names of authors in the paragraph

If not: give reasons

[REPLY] Thank you for comment. We added a column “other outcomes” in Table 1 summarizing all other outcomes mentioned within each individual included study. We also rephrased the section as following, including the list of references (studies) for each outcome:

“Metabolic control: (HbA1c) was reported in all studies except two [22,24]. The mean HbA1c patients following exercise training was 7.5 % (95% CI 7.2 to 7.7) in T2DM patients, ranging from 10.4 ±2.2 [64] to 6.4 ±0.6 [54], and 7.7 % (95% CI 7.1 to 8.4) in controls, ranging from 8.7 ±0.32 [52] to 6.4 ±0.5 [54]. 

BMI was reported in all studies except four [19,24,59]. The mean BMI patients following exercise training was 29.5 kg/m² (95% CI 28.3 to 30.7) in T2DM patients, ranging from 39.2 ±9.4 [53] to 23.9 ±2.9 [54], and 28.3 kg/m² (95% CI 27.1 to 29.6) in controls, ranging from 34.6 ±1.8 [56] to 25.5 ±3.1 [54].

Blood pressure was reported in all studies except seven [19,20,23,53,56,59,61]. Mean blood pressure (systolic/diastolic) following exercise training was 129.2/81.2 mmHg (95% CI 123.0/78.7 to 135.3/83.7) in T2DM patients, ranging from 144.2/88.6 [52] to 117.3 [62] / 61.5 [60] and 133.2/79.6 mmHg (95% CI 119.8/73.0 to 146.7/86.1) in controls group without exercise, ranging from 145.2/87.0 [52] to 119.0 [25] / 70.0 [58].

Blood lipid levels: total cholesterol was reported in 11 studies [20,23,54–56,58–60,62–64] HDL cholesterol in 13 studies [20,23,53–56,58–60,62,63,65,67], LDL in 11 [20,23,54,56,58–60,62–65] and triglycerides in 12 studies [20,23,53,54,56,59,60,62–65,68].

Aerobic capacity: VO2max and VO2peak were reported in 3 [53,54,57] and 10 studies [20,21,23,25,55,56,59–62]. Mean VO2max/peak was 24.3 mL.min-1.kg-1 (95% CI 22.3 to 26.2) for exercise groups before training program, ranging from 18 ±2.8 mL.min-1.kg-1 [59] to 31.9 ±5.1 [57], and 25.2 mL.min-1.kg-1 (95% CI 20.5 to 29.9) for controls. ranging from 20.3 ±1.8 [25] to 32.2 ±6.4 [57].”

Reviewer 3

Good piece of work and very good effort from the authors in collecting the data. However, there are some certain points need to be corrected or explained before considering this manuscript for publication.

[REPLY] Thank you for positive comment. We have addressed your concern below.

Introduction:

There should be a paragraph explaining the HRV giving what time domain include and what frequency domain include.

[REPLY] Thank you for comment. As the introduction is already long, we added the following summary of HRV in the Introduction: “The HRV analysis can provided detailed information about cardiac regulatory system and it has been demonstrated that T2DM patients exhibited a strong decrease in HRV [5,6]. HRV is basically the variation between two consecutive heartbeats (RR-intervals) [9]. HRV can be analyzed through various parameters, classically classified as time and frequency domains. Time domains are calculation from RR-intervals (time between two heartbeats), and frequency domains are a more complex power spectral analysis of the HRV. Both domains comprised several parameters informing on the activity of the autonomic nervous system, such as sympathetic or parasympathetic activity [10].” We gave details on parameters within both time domain and frequency domain in the Methods section: “The primary outcome analysed was HRV parameters. Time-domain parameters were RR–intervals (or Normal to Normal intervals – NN), standard deviation of RR intervals (SDNN), percentage of adjacent NN intervals varying by more than 50 milliseconds (pNN50), and root mean square of successive RR-intervals differences (RMSSD). Frequency-domain parameters were total power (TP), low frequency (LF), high frequency (HF) and LF/HF ratio. The RMSSD and pNN50 are associated with HF power and hence parasympathetic activity, whereas SDNN is correlated with LF power. Even if LF power is an index of both sympathetic and parasympathetic activity, LF power is commonly considered as a measure of sympathetic modulations, particularly when expressed in normalised units. In practical terms, an increase of the LF component is generally considered to be a consequence of an increased sympathetic activity [38]. HF power represents the most efferent vagal (parasympathetic) activity to the sinus node [8,39–42]. Therefore, an increase of the HF component reflects an increased parasympathetic activity. The LF/HF ratio represents the sympathovagal balance (Table 1).”

Also there are many methods to assess the cardiac autonomic function, why did you specify it only to HRV?

[REPLY] Thank you for comment. We added a whole section on the interest of using HRV parameters to assess HRV. HRV is free, easily accessible, and non-intrusive, instantly measured, pain-free. In particular, the introduction now reads: “Despite the gold standard to assess CAN is still cardiovascular reflex tests [16], one of the most convenient and reliable assessment is through HRV. HRV can be measured easily using a portable device, non–intrusively and pain–free [19].”

The authors mentioned that studies in HRV in T2DM are scarce, give an explanation and an example to those studies. 

[REPLY] Thank you for comment. Yes, this is the subject of our meta-analysis. Few studies assessed HRV in T2DM, and there were still many unanswered before our meta-analysis. We added several sentences to show the readers what is known and what is not known, and the need for our meta-analysis. The introduction now reads: “Exercise training is a cornerstone of lifestyle intervention [6–8], leading to improved HRV in healthy population [9], but it remains unclear to what extent physical exercise can improve HRV in T2DM. In T2DM, different modalities of exercise have been tested such as endurance [10–12], resistance [13,14], or high intensity interval training [15,16]. HIIT provides greater benefits to functional capacity comparing to endurance training (11). Resistance training similarly to endurance training, improves metabolic features, insulin sensitivity and reduces abdominal fat (12,13). However, benefits on HRV depending on exercise modality remain unclear. Moreover, other characteristics of training may influence the results. For example, supervised exercises has been proved more effective than non-supervised on several outcomes [17–19], including in T2DM. Similarly, duration and frequency of training, are strongly linked with putative benefits, but evidence is scarce on HRV in T2DM. Characteristics of patients can also influence benefits of exercise on HRV [20,21]. Lastly, the relationships between changes in HRV and clinical or biological parameters has also been poorly studied [22–24].”

Study designs and objectives: the 21 included studies. please explain which one of them used frequency domain or time domain.

[REPLY] Thank you for comment. The details of studies are presented in the Results section devoted to “HRV parameters” that reads: “For time domain parameters, RR intervals (RRI)was reported in 5 studies [16,44,46,52,55], SDNN in 13 studies [13,15,16,39–41,44,46,49–51,53,55], RMSSD in 9 studies [11,13,39,41,46,49–51,55], PNN50 in 6 studies [13,40,49–51,53]. For frequency domain parameters, the total power was reported in 6 studies [14,43,47,48,52,53], LF in 13 studies [11,12,14,16,39–41,46,49–52,55], HF in 13 studies [11,12,14,16,39–41,43,46,49–52] and LF/HF in 12 studies [12,14,16,39–41,43,46,49–52]. We excluded inconsistent data of LF/HF from one study [47].”

Inclusion and exclusion criteria section: it was stated "In most studies, sedentary behavior or low level of physical activity was necessary" why it was necessary, please give full explanation

[REPLY] Thank you for comment. As the aim of studies was to assess the effect of exercise, they included patients who did not exercise. The sentence now reads: “Most studies included patients with sedentary behavior or low level of physical activity […]”

Metabolic control section: "the mean HbA1c in T2DM patients following exercise training was 7.5 %" The percentage here after 7.5 has no meaning.

[REPLY] Thank you for comment. Sorry, we do not understand the question. The common unit of HbA1c is a percentage (%). Please see for example 

Aerobic capacity section: Please identify if VO2 max was measured or Vo2 peak, as in some area of the manuscript Vo2 peak was mentioned. Considering the age of the participants, I doubt that Vo2 max was measured, but please check.

[REPLY] Thank you for comment. The sentence now reads: “VO2max and VO2peak were reported in 3 [53,54,57] and 10 studies [20,21,23,25,55,56,59–62].”

Duration of measures: The validity of HRV less than 2 min is questioned, therefore, I am concerned how recording of HRV of 1 min was included. Look for the task force of the European Society of Cardiology and the North American Society of Pacing and Electrophysiology. The recommendation for short recording are only valid and specified for frequency domain analysis.

[REPLY] Thank you for comment. We totally agree with you. In fact, those articles were not includued in the meta-analyses because they were the only one reporting a non-consensual measure of HRV, without reporting other HRV parameters. Computing meta-analyses on less than five articles is very debatable. Moreover, they also measured HRV manually from a very short period (1 minute) which is not recommended. The methods Statistical considerations section now reads: “We conducted random effects meta–analyses (DerSimonian and Laird approach) when data could be pooled (more than five data for the same outcome) [37]. A particular attention was paid for short recording (1 minute) of HRV parameters (REF: Task force of the European Society of Cardiology and the North American Society of Pacing and Electrophysiology).” The results section now reads: ““The use of the selection criteria reduced the number of articles reporting the effect of exercise on HRV in T2DM patients to 21 articles in the systematic review, among which 18 articles were included in the meta-analysis (inter-reader agreement κ = 0.89) (Fig 1). The three articles were not included in the meta-analysis because they only reported ratio between longest and shortest RR-intervals; they were also distinguishable because they measured HRV manually on electrocardiogram recording of 1 minute [10,42,45].” The table now includes the following legend: “†: articles not included in the meta-analyses.”

Other metaregressions section:

Please report the p value as well as the unit of measurement whether it is ms or normailzed unit.

[REPLY] Thank you for comment. We added all p-values in the section. The Methods “Statistical considerations” section now reads: “As ES is a unitless measure and as we compared data after and before exercise, frequency-domain HRV parameters measured in ms2 or in normalized unit (nu) were combined.” Therefore, the “Other metaregressions” section has no unit for LF and HF. Results were similar whether we distinguished LF and HF in ms2 or in normalized unit. In order to be the more concise we did not spontaneously present those results. We already have fifteen supplementary materials. However, we can provide four additional supplementary materials if needed (LF ms2, HF ms2, LF nu, HF nu).

Also I suggest adding creating a table where the meaning of LF and HF can be easily tracked. For example does LF measures purely sympathetic.

Also when improvement in LF is mentioned, does that indicate a decrease or increase?

[REPLY] Thank you for comment. We added a new “Table 1. Descriptive characteristics of HRV parameters” summarizing the meaning of all HRV parameters. In addition to the new Table 1, we also added the following sentences within the Methods section: “Even if LF power is an index of both sympathetic and parasympathetic activity, LF power is commonly considered as a measure of sympathetic modulations, particularly when expressed in normalised units. In practical terms, an increase of the LF component is generally considered to be a consequence of an increased sympathetic activity (REF). HF power represents the most efferent vagal (parasympathetic) activity to the sinus node [8,38–41]. Therefore, an increase of the HF component reflects an increased parasympathetic activity […] (Table 1).” We added the following reference: “Sztajzel J. Heart rate variability: a noninvasive electrocardiographic method to measure the autonomic nervous system. Swiss Med Wkly. 2004 Sep 4;134(35-36):514-22”.

In the first paragraph of the discussion, define what is HRV improvement?

[REPLY] Thank you for comment. The first paragraph of the discussion now reads: “The main findings were that exercise training improved HRV in T2DM patients, with a decrease in sympathetic activity and an increase in parasympathetic activity.”

Which parameter in HRV represent the improvement in sympathovagal balance

[REPLY] Thank you for comment. We added a new “Table 1. Descriptive characteristics of HRV parameters” summarizing the meaning of all HRV parameters. In addition to the new Table 1, we also added the following sentences within the Methods section: “The LF/HF ratio represents the sympathovagal balance”.

"We also showed that exercise improved less HRV in T2DM patients reporting the use of metformin" This statement is not clear. please clarify.

[REPLY] Thank you for comment. The sentence now reads: “T2DM patients using metformin improved less their HRV after exercise compared with T2M patients that did not use metformin.”

Also I suggest instead of using decreased HRV, poor HRV.

[REPLY] Amended.

"Few effects on lipid level profile have been demonstrated and it remains unclear to what extent changes in blood lipids contribute to the cardiovascular benefits of exercise" This statement also need clarification.

[REPLY] Thank you for comment. The sentence now reads: “For example, it remains unclear to what extent changes in blood lipids contribute to the cardiovascular benefits of exercise [16,88].”

Overall, there should be paragraph indicating the effect of different positions while measuring HRV. 

[REPLY] Thank you for comment. The Limitations section now reads: “To reduce bias of measures, when a study reported HRV in different positions (REF), we limited data to decubitus measures, as position and conditions of measure may influence HRV.” We added the following reference: “da Cruz CJG, Porto LGG, da Silva Rolim P, de Souza Pires D, Garcia GL, Molina GE. Impact of heart rate on reproducibility of heart rate variability analysis in the supine and standing positions in healthy men. Clinics (Sao Paulo). 2019;74:e806. doi: 10.6061/clinics/2019/e806.”

Also The effect of different medication especially beta blockers on HRV.

[REPLY] Thank you for comment. We added the following sentence within the discussion: “Betablockers are known to affect HRV (REF). We cannot conclude that betablockers influenced response in HRV to exercise, as only one study reported its use [58] and betablockers being explicitly an exclusion criteria in most studies [19,21,23,25,52,55,56,59–61,63,64].” The following reference was added: “Elghozi JL, Girard A, Laude D. Effects of drugs on the autonomic control of short-term heart rate variability. Auton Neurosci 2001 Jul 20;90(1-2):116-21. doi: 10.1016/S1566-0702(01)00276-4”. 

In the conclusion: it is mentioned endurance, and high intensity interval, please clarify what is the difference?

[REPLY] Thank you for comment. The sentence now reads: “The level of proof is the highest for endurance training (aerobic), despite resistance (anaerobic) and high-intensity-interval training (alternating short intense anaerobic and less intense exercises) may also be promising.”

We hope our work will be considered favorably and look forward to hearing from you.

---

## [Decision Letter · Decision Letter 1]

11 Apr 2021

PONE-D-20-40085R1

Effect of exercise training on heart rate variability in type 2 diabetes mellitus patients: A systematic review and meta-analysis

PLOS ONE

Dear Dr. Navel,

Thank you for submitting your manuscript to PLOS ONE. After careful consideration, we feel that it has merit but does not fully meet PLOS ONE’s publication criteria as it currently stands. Therefore, we invite you to submit a revised version of the manuscript that addresses the points raised during the review process.

We look forward to receiving your revised manuscript.

Kind regards,

Walid Kamal Abdelbasset, Ph.D.

Academic Editor

PLOS ONE

Journal Requirements:

Reviewers' comments:

Reviewer's Responses to Questions

**Comments to the Author**

1. If the authors have adequately addressed your comments raised in a previous round of review and you feel that this manuscript is now acceptable for publication, you may indicate that here to bypass the “Comments to the Author” section, enter your conflict of interest statement in the “Confidential to Editor” section, and submit your "Accept" recommendation.

Reviewer #1: All comments have been addressed

Reviewer #2: All comments have been addressed

Reviewer #3: All comments have been addressed

2. Is the manuscript technically sound, and do the data support the conclusions?

Reviewer #1: Yes

Reviewer #2: Yes

Reviewer #3: Yes

3. Has the statistical analysis been performed appropriately and rigorously? 

Reviewer #1: Yes

Reviewer #2: Yes

Reviewer #3: Yes

4. Have the authors made all data underlying the findings in their manuscript fully available?

Reviewer #1: Yes

Reviewer #2: Yes

Reviewer #3: Yes

5. Is the manuscript presented in an intelligible fashion and written in standard English?

Reviewer #1: Yes

Reviewer #2: Yes

Reviewer #3: Yes

6. Review Comments to the Author

Reviewer #1: The authors have satisfacrily answered all the comments raised by me and can be published in presnt format.

Reviewer #2: thanks alot for your fast and perfect replay for all comments of the reviewers

and thank you for your idea

Reviewer #3: The common unit for HbA1c reported in research is mmol/mol. If the data extracted from the literature is in %, then it is fine.

In the aerobic section: VO2max/VO2peak are not the same when written in this way it implies that they are when they are not. Please make sure to report the studies used VO2 peak and those used VO2 max and report whether they were extracted via direct measure or estimation.

Finally, the introduction part, I suggest dividing your paragraphs instead of having all the information in one go. For example, in the intro, the manuscript introduce DM and how it can lead to comorbidities and CAN. Then a second pragraph should follow how CAN can be measured. Then how exercise / lifestyle modification contribute to the HRV in dm. This will help the reader to comprehend the flow of information provided in the intro.

7. PLOS authors have the option to publish the peer review history of their article (what does this mean?). If published, this will include your full peer review and any attached files.

Reviewer #1: **Yes: **Gopal Nambi

Reviewer #2: No

Reviewer #3: No

---

## [Decision Letter · Decision Letter 2]

5 May 2021

Effect of exercise training on heart rate variability in type 2 diabetes mellitus patients: A systematic review and meta-analysis

PONE-D-20-40085R2

Dear Dr. Navel,

We’re pleased to inform you that your manuscript has been judged scientifically suitable for publication and will be formally accepted for publication once it meets all outstanding technical requirements.

Kind regards,

Walid Kamal Abdelbasset, Ph.D.

Academic Editor

PLOS ONE

Additional Editor Comments (optional):

Reviewers' comments:

Reviewer's Responses to Questions

**Comments to the Author**

1. If the authors have adequately addressed your comments raised in a previous round of review and you feel that this manuscript is now acceptable for publication, you may indicate that here to bypass the “Comments to the Author” section, enter your conflict of interest statement in the “Confidential to Editor” section, and submit your "Accept" recommendation.

Reviewer #3: All comments have been addressed

2. Is the manuscript technically sound, and do the data support the conclusions?

Reviewer #3: (No Response)

3. Has the statistical analysis been performed appropriately and rigorously? 

Reviewer #3: (No Response)

4. Have the authors made all data underlying the findings in their manuscript fully available?

Reviewer #3: (No Response)

5. Is the manuscript presented in an intelligible fashion and written in standard English?

Reviewer #3: (No Response)

6. Review Comments to the Author

Reviewer #3: (No Response)

7. PLOS authors have the option to publish the peer review history of their article (what does this mean?). If published, this will include your full peer review and any attached files.

Reviewer #3: No

---

## [Editor Report · Acceptance letter]

7 May 2021

PONE-D-20-40085R2 

Effect of exercise training on heart rate variability in type 2 diabetes mellitus patients: A systematic review and meta-analysis 

Dear Dr. Navel:

I'm pleased to inform you that your manuscript has been deemed suitable for publication in PLOS ONE. Congratulations! Your manuscript is now with our production department. 

Kind regards, 

on behalf of

Dr. Walid Kamal Abdelbasset 

Academic Editor

PLOS ONE